# CoPoP liposomes displaying stabilized clade C HIV-1 Env elicit tier 2 multiclade neutralization in rabbits

Annemart Koornneef[1,5], Kanika Vanshylla[1,5], Gijs Hardenberg[1], Lucy Rutten [1], Nika M. Strokappe [1], Jeroen Tolboom[1], Jessica Vreugdenhil[1], Karin Feddes-de Boer[1], Aditya Perkasa[1], Sven Blokland [1], Judith A. Burger[2], Wei-Chiao Huang[3], Jonathan F. Lovell[3], Danielle van Manen[1], Rogier W. Sanders [2], Roland C. Zahn [1], Hanneke Schuitemaker [1], Johannes P. M. Langedijk[1,4] ✉ & Frank Wegmann [1] ✉

One of the strategies towards an effective HIV-1 vaccine is to elicit broadly neutralizing antibody responses that target the high HIV-1 Env diversity. Here, we present an HIV-1 vaccine candidate that consists of cobalt porphyrin-phospholipid (CoPoP) liposomes decorated with repaired and stabilized clade C HIV-1 Env trimers in a prefusion conformation. These particles exhibit high HIV-1 Env trimer decoration, serum stability and bind broadly neutralizing antibodies. Three sequential immunizations of female rabbits with CoPoP liposomes displaying a different clade C HIV-1 gp140 trimer at each dosing generate high HIV-1 Env-specific antibody responses. Additionally, serum neutralization is detectable against 18 of 20 multiclade tier 2 HIV-1 strains. Furthermore, the peak antibody titers induced by CoPoP liposomes can be recalled by subsequent heterologous immunization with Ad26-encoded membrane-bound stabilized Env antigens. Hence, a CoPoP liposome-based HIV-1 vaccine that can generate cross-clade neutralizing antibody immunity could potentially be a component of an efficacious HIV-1 vaccine.

The need remains for an HIV-1 vaccine that can provide sterilizing immunity to manage the ongoing global HIV pandemic despite successes in antiretroviral treatment and pre-exposure prophylaxis[1,2]. A major challenge to the HIV-1 vaccine field is the high antigenic diversity[3] and the metastable nature of the viral envelope glycoprotein (Env)[4] which is the only protein exposed on the viral surface and hence key to blocking viral entry and establishment of infection. To effectively tackle the virus, an HIV-1 vaccine should elicit both potent and broad immunity against the Env protein.

A multitude of approaches have been used to develop HIV-1 vaccine candidates that can trigger antibody and/or T cell immunity. These include the use of Env as an immunogen, as well as structural proteins like Gag, Pol and Nef being delivered as adjuvanted proteins[5], DNA-encoded[6,7], canarypox-vectored[8], Ad5-vectored antigens[6], or Ad26-vectored mosaic immunogens[9]. Despite promising immunogenicity and protective efficacy in animal models, phase III trials were disappointing with limited to no efficacy seen in humans[10,11]. These HIV-1 vaccines did not aim at eliciting broadly neutralizing antibodies (bNAbs) that target multiple HIV-1 strains, including widely circulating, difficult-to-neutralize tier 2 strains. HIV-1 bNAbs are a rare class of antibodies identified in and isolated from people living with HIV-1[12] that can bind conserved epitopes on the Env glycoprotein.

[1]Janssen Vaccines & Prevention, Leiden, The Netherlands. [2]Department of Medical Microbiology, Amsterdam UMC, University of Amsterdam, Amsterdam Institute for Infection and Immunity, Amsterdam, the Netherlands. [3]Department of Biomedical Engineering, University at Buffalo, Buffalo, NY, USA. [4]Present address: ForgeBio, Amsterdam, The Netherlands. [5]These authors contributed equally: Annemart Koornneef, Kanika Vanshylla. ✉e-mail: hlangedijk@forge-bio.com; frank.wegmann.1@gmail.com

Administration of bNAbs was demonstrated to provide protection against single[13,14] or repeated challenges[15] of chimeric simian-human immunodeficiency virus (SHIV) in non-human primates (NHPs). Building on that, the antibody-mediated prevention (AMP) trials assessed the protective efficacy of the VRC01 bNAb. Despite lack of overall efficacy, this was the first study to demonstrate that protection from bNAb-sensitive strains is a possibility in humans[16]. Considering the promising nature of bNAbs, and with the goal to elicit them by activating rare B cell lineages, novel HIV-1 vaccine candidates are in early development[11]. In one such approach, the germline-targeting eOD-GT8 immunogen could activate VRC01-class B cell precursors in a phase I human trial. As no neutralization was observed in that case, protective neutralizing antibody responses would likely require subsequent immunizations to drive maturation of the early B cell lineages[17].

The high antigenic diversity of Env, rare occurrence of bNAb-producing B cell precursors and low affinity of the germline B cell receptors to Env point towards the need for sequential immunization with multiple Env antigens to gradually guide the immune system towards bNAb responses[18,19]. Stable HIV-1 Env trimer immunogens that closely mimic the prefusion closed Env conformation on the virus have been recently developed[20,21]. We established a universal repair and stabilize (RnS) approach that stabilizes HIV-1 Env with a selection of amino acid substitutions in highly unstable domains of gp41 centered around the hybrid sheet, and replaces rare amino acids with consensus residues, thereby improving quality and yield of the protein[22,23]. However, soluble Env trimers can present immunodominant non-neutralizing epitopes at the trimer base[24,25], an issue that could be overcome by delivery of a membrane-bound antigen, or by presenting HIV-1 Env on a particle. Transgene-mediated expression of membrane-bound Env trimers has the benefit of a glycosylation profile that closely matches Env on a virion[26], while presentation of recombinant Env on a particle allows purification of correctly folded trimer antigens prior to coupling. In addition, display of Env on a particle increases local antigen density and facilitates activation of low affinity germline B cells[27].

Particle presentation of Env has been tested in animal studies in various forms including self-assembling nanoparticles[17,28] and virus-like particles (VLPs)[29,30]. Although these platforms are promising, properties like off-target responses to protein nanoparticles or low trimer decoration and the display of non-neutralizing Env epitopes on VLPs may hamper generation of HIV-1 bNAb responses[31,32]. Liposomes provide a versatile, biocompatible platform to deliver protein antigens as well as to co-formulate lipid adjuvants[33–35]. Presentation of histidine-tagged (His-tagged) HIV-1 Env on nickel nitrilotriacetic acid (Ni-NTA)-functionalized liposomes has been described, but the interaction between Env and Ni-NTA readily dissociates in serum and hence these liposomes are less suitable for in vivo application[27,36–38]. In contrast, an emerging technology involves cobalt porphyrin-phospholipid (CoPoP) liposomes, which localizes cobalt in the hydrophobic bilayer sheltered from water, thereby increasing the stability of the His-tag protein anchor relative to the Ni-NTA lipid approach[39,40]. Thus, CoPoP can easily directionally and biostably complex His-tagged antigens via cobalt in the lipid bilayer. This strategy was demonstrated to be safe and immunogenic in humans in the context of a SARS-CoV-2 vaccine candidate[41], but has not yet been tested in the delivery of stabilized HIV-1 Env trimer immunogens.

With the aim to elicit broadly neutralizing antibody responses, we here combined the use of CoPoP liposomes and stabilized HIV-1 Env proteins to generate a particle-based delivery platform to present HIV-1 Env trimers in a prefusion conformation to B cells in vivo. Three different CoPoP liposome formulations decorated with HIV-1 Env trimers were tested for their in vitro stability, Env trimer decoration and antigenicity. To study immunogenicity, rabbits were immunized sequentially with CoPoP liposomes decorated with three different clade C HIV-1 Env trimers, and Env binding antibodies as well as serum neutralizing antibody responses were assessed. In addition, liposome immunizations were followed by dosing with Ad26-encoded HIV-1 Env immunogens to assess recall responses with a heterologous vaccine platform.

## Results

### CoPoP liposomes decorated with stabilized HIV-1 Env antigens exhibit high Env trimer decoration, serum stability and bNAb antigenicity

To obtain high quality prefusion closed trimers, we applied the RnS method to a consensus clade C (ConC) sequence, a mosaic clade C (sC4) sequence based on a compilation of Env sequences with a broad coverage of T cell epitopes[42], and to a tier 2 clade C strain, *C97ZA*.012 (C97ZA) (Supplementary Fig. 1)[22]. Quality and stability of the purified trimers was determined by biolayer interferometry using Octet (Fig. 1a) and by measuring the melting temperature ($Tm_{50}$) by nano Differential Scanning Fluorimetry (nanoDSF), respectively (Fig. 1b). A panel of HIV-1 anti-gp120 antibodies against major antigenic sites including the V1/V2 region (PGT145, VRC26, PGDM1400), the V3 base glycan supersite (2G12, PGT128), the CD4 binding site (CD4bs) (VRC01, 3BNC60), the fusion peptide (VRC34, PGT151), and the gp120/gp41 interface (35O22) was assessed[43] (Fig. 1a). The binding pattern was reflective of a prefusion closed HIV-1 Env conformation, showing bNAb binding in the absence of non-bNAb binding. As a comparator, a consensus clade B (ConB) protein that lacks the V1/V2 binding epitope for PGT145, VRC26, and PGDM1400 was used (Supplementary Fig. 1). ConB showed a more open conformation, as demonstrated by binding of non-bNAbs. In concordance, ConB reached a $Tm_{50}$ of 66.6 °C, while stabilized clade C HIV-1 Envs showed enhanced thermostability with melting temperatures ranging between 73.4 °C and 80.3 °C.

To present a dense array of HIV-1 Env trimers and to mask immunodominant, non-neutralizing cryptic epitopes at the base of the soluble Env trimer, we directionally complexed each of the stabilized clade C HIV-1 Env trimers individually to CoPoP liposomes. Three different liposomal formulations, CoP1, CoP2, and CoP3, were assessed that each differ in CoPoP content, phospholipid composition and adjuvant incorporation (Fig. 1c). The average HIV-1 Env trimer decoration of three clade C HIV-1 Env-CoPoP formulations was approximately 88-147 Env trimers per liposome (Fig. 1c). Env-CoPoP liposome batches for in vivo application were purified by size exclusion chromatography (SEC) to remove unbound Env (Supplementary Fig. 2). The collected Env-CoPoP liposome fraction was assessed by dynamic light scattering (DLS) and the diameter and polydispersity index (PDI) was compared to liposomes without HIV-1 Env decoration (Fig. 1c). The average diameter of Env-decorated CoPoP liposomes was approximately 116 nm with a PDI below 0.1, indicative of a monodisperse sample. For CoP1 and CoP2 liposomes, there was a 20-25 nm diameter increase when HIV-1 Env trimers were present on liposomes. In contrast, there was no clear diameter increase for CoP3 liposomes, even though HIV-1 Env decoration was confirmed by SEC and antigenicity assays (Supplementary Fig. 2, Fig. 1e). Purified Env-CoPoP liposomes were subjected to cryo electron microscopy (cryo-EM) analysis, confirming the presence of a dense array of spikes on the surface of the liposomes (Fig. 1d).

The antigenicity of HIV-1 Env trimers decorating CoPoP liposomes and of soluble ConC and ConB Env trimers was assessed by an amplified luminescent proximity homogeneous assay (AlphaLISA) (Fig. 1e). This bead-based luminescence amplification assay yields a signal when antibody-labeled donor and acceptor beads are in close proximity. PGT128 bNAb was used as both a biotinylated (PGT128-biotin) and a V5-tag (PGT128-V5) labeled binding partner, demonstrating comparable binding profiles across CoP1, CoP2, and CoP3 formulations for each clade C HIV-1 Env trimer. V1/V2 apex directed bNAb PGT145 was also used as both a biotinylated (PGT145-biotin) and a V5-tag (PGT145-V5) labeled binding partner. This PGT145 AlphaLISA

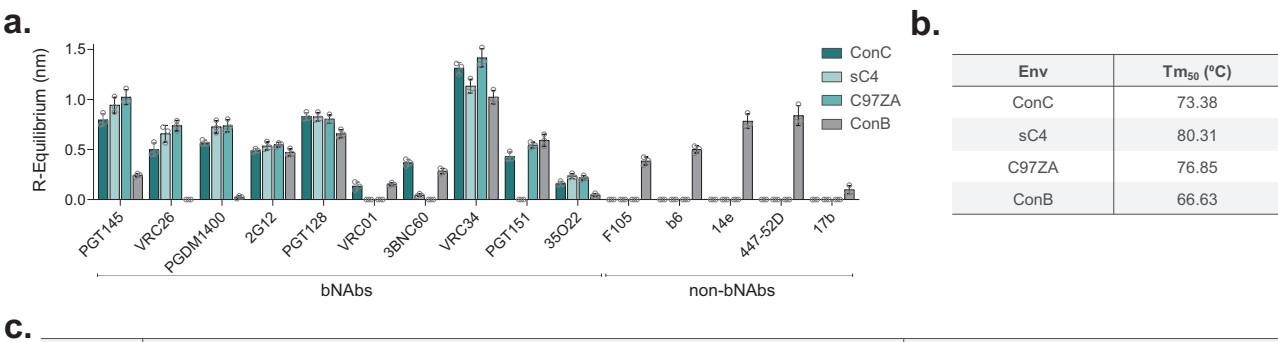

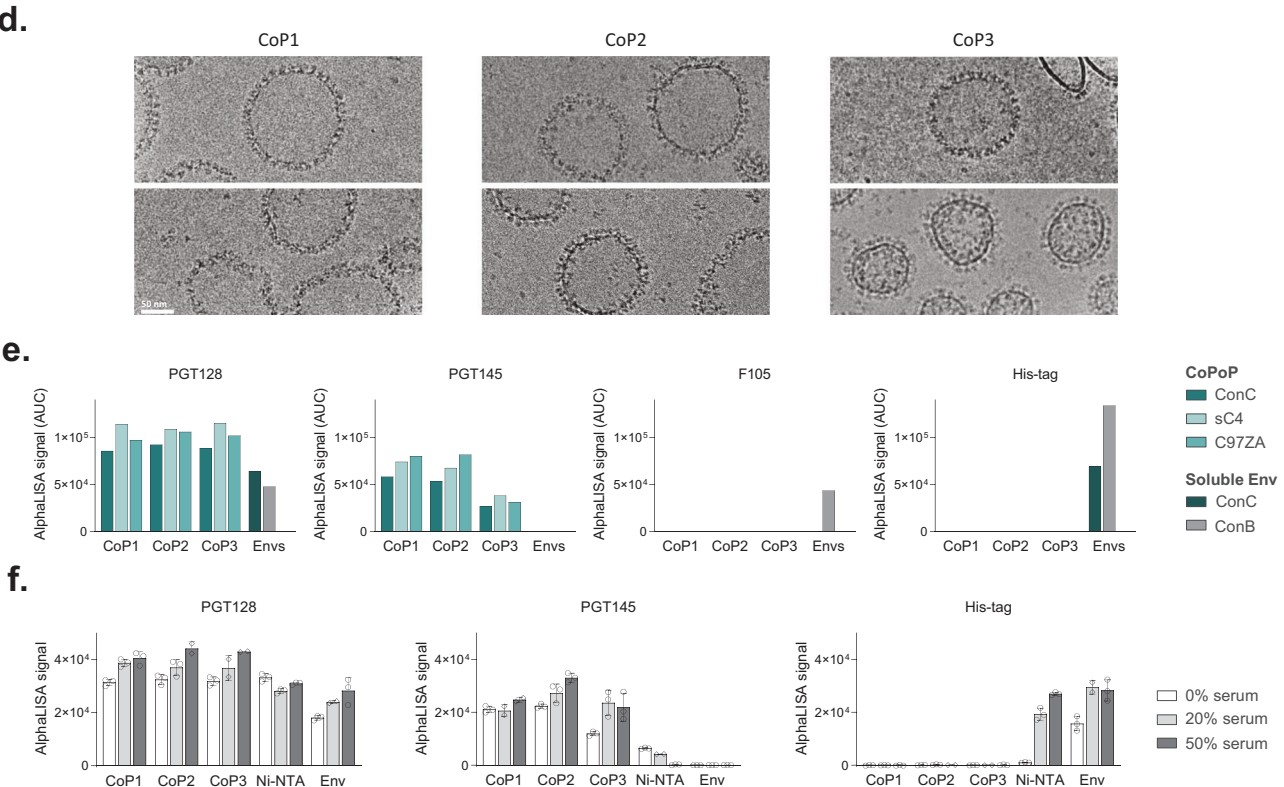

**Fig. 1 | In vitro characterization of CoPoP liposomes decorated with stabilized HIV-1 Env antigens. a** Antigenic binding profile of soluble His-tagged clade C HIV-1 Envs ConC, sC4, and C97ZA and clade B ConB Env against a panel of broadly neutralizing antibodies (bNAbs) (PGT145, VRC26, PGDM1400, 2G12, PGT128, VRC01, 3BNC60, VRC34, PGT151 and 35O22) and non-bNAbs (F105, b6, 14e, 447-52D and 17b) using biolayer interferometry. Bars represent average ± standard deviation (SD) of $n = 3$ technical replicates (open circles). **b** Melting temperature ($Tm_{50}$) of HIV-1 Env trimers from **a** as measured by nano differential scanning fluorimetry (nanoDSF). **c** Composition, diameter, and polydispersity index (PDI) of CoPoP liposome formulations CoP1, CoP2, and CoP3. Average diameter, PDI, and HIV-1 Env trimer density of CoPoP liposomes decorated with clade C HIV-1 Env trimers ConC, sC4 or C97ZA. The use of DOPC phospholipid is indicated with *. **d** Cryo-Electron Microscopy (Cryo-EM) images of CoP1, CoP2, and CoP3 liposomes decorated with sC4 Env. Scale bar = 50 nm. Two representative images per grid are shown from a single Cryo-EM analysis experiment. **e** Antigenic binding profile of ConC, sC4, and C97ZA HIV-1 Env trimers displayed on CoPoP liposomes (CoP1, CoP2, CoP3) measured by AlphaLISA binding assay, using bNAbs PGT128 and PGT145, non-bNAb F105, and by anti-His tag detection. Soluble His-tagged ConC and ConB proteins (Envs) were included as controls. Bars represent area under the curve of AlphaLISA signal obtained with a dilution series of Env. **f** Rabbit serum stability of BG505-CoPoP (CoP1, CoP2, CoP3) and BG505-Ni-NTA (Ni-NTA) liposomes measured by AlphaLISA binding assay using bNAbs PGT128 and PGT145 and by anti-His tag detection. Soluble His-tagged BG505 protein (Env) was included as a control. Bars represent average AlphaLISA signal ± standard deviation (SD) of $n = 2$ (open diamonds) or $n = 3$ (open circles) technical replicates. Source data are provided as a Source Data file.

set-up only yields a signal when multiple trimers are in proximity, since PGT145 binds a single epitope at the apex of one HIV-1 Env trimer. Accordingly, no binding to soluble HIV-1 Env was detected, while binding of PGT145 to Env-CoPoP liposomes was observed. PGT145 binding to Env-CoP3 particles was lower compared to Env-CoP1 and Env-CoP2 liposomes. Finally, the absence of binding to non-bNAb F105 was confirmed by employing PGT128-V5 together with F105-biotin. The absence of unbound His-tagged HIV-1 Env was confirmed in AlphaLISA by combining anti-His acceptor beads with PGT128-V5 (Fig. 1e).

To test the stability of Env-CoPoP liposomes, the particles were incubated with rabbit serum and assessed by AlphaLISA (Fig. 1f). As a control, Env-Ni-NTA liposomes that have been shown to dissociate in serum were prepared in a similar fashion to CoPoP liposomes. All liposomes for the serum stability study were decorated with His-tagged stabilized BG505 HIV-1 Env trimers[22] (Supplementary Fig. 1). The His-tag became readily detectable for HIV-1 Env-decorated Ni-NTA liposomes upon incubation with increasing serum concentrations, indicating release of Env from liposomes. In addition, there was a marked reduction in PGT145 signal for HIV-1 Env-decorated Ni-NTA liposomes in comparison with controls. In contrast, HIV-1 Env remained stably attached to CoPoP liposomes in the presence of serum, in agreement with previous observations[40].

### HIV-1 Env-CoPoP liposomes induce antigen-specific antibody responses in rabbits

In order to study the immunogenicity of HIV-1 Env-decorated CoPoP liposomes, New Zealand White (NZW) rabbits were used, as they can produce antibodies with long heavy chain complementarity determining region 3 (HCDR3s) which are common to many bNAbs[44]. The comparison of the three different liposome compositions was done by immunizing up to 9 animals per group with either CoP1, CoP2 or CoP3 liposomes. Each liposome type was decorated with one of three HIV-1 clade C Envs and used for sequential immunization, with ConC-CoPoP, sC4-CoPoP and C97ZA-CoPoP at week 0, week 8 and week 16 respectively (Fig. 2a). The animals were dosed via the intramuscular (i.m.) route with 30 µg of the respective Env antigen. With the liposome compositions used here, the amount of adjuvant incorporated into the particles is low compared to earlier rabbit studies[45]. Therefore, to maximize immunogenic potential, all groups were adjuvanted with Adjuplex[46,47] at each immunization time point. With the aim to elicit broad responses against conserved epitopes, the clade C consensus sequence-based ConC was used as the first immunogen, followed by sC4 and C97ZA. The rationale was to mimic the progressive diversification of transmitter/founder viruses in infected individuals that are able to generate bNAb responses over time. In that regard, ConC is most, and C97ZA least homologous to transmitter/founder viruses at key Env amino acid positions that bind bNAbs[48–51]. A fourth group of control animals (N = 4) received buffer at each immunization (Fig. 2a).

Two weeks after the third immunization (week 18), sera were used to assess the immunogenicity of this three-dose regimen. Total binding antibody levels against the ConC gp140 trimer were first tested in an ELISA where Env was directly coated on to the ELISA plates (Fig. 2b). The three groups that received the different HIV-1 Env liposome formulations displayed comparable binding antibody responses with median anti-HIV-1 titers of 4.39, 4.43 and 4.29 log EU/mL for groups Env-CoP1, Env-CoP2 and Env-CoP3, respectively (Fig. 2b; Supplementary Table 1). Capturing the HIV-1 Env antigen via a tag maintains conformational epitopes that might be destroyed when directly coating Env on the surface of the ELISA plate. Therefore, we used the C-terminal biotin-tag on ConC gp140 trimers to capture Env on Streptavidin-coated plates and hence maintained a more native-like trimer conformation in a capture coat ELISA. The antibody response to conformational epitopes was similar for all three groups, with titers of 4.14, 4.17 and 4.21 EU/mL for groups Env-CoP1, Env-CoP2 and Env-

CoP3, respectively (Fig. 2c; Supplementary Table 1). Sera from all control animals remained below detection limit in both ELISA formats, confirming antigen specific serum responses (Fig. 2b, c).

Since the Env proteins used here had a terminal His-tag for attachment to the CoPoP liposome bilayer, high His-specific antibody levels would indicate detachment of HIV-1 Env from the liposome. CoPoP liposomes were previously shown to have better retention of antigens on the particle compared to a Ni-NTA-based delivery system[40]. Our own analyses from a rabbit study where we compared Ni-NTA to CoP1 liposomes for HIV-1 Env delivery in a similar immunization schedule (Supplementary Fig. 3a) and with comparable gp140 responses (Supplementary Fig. 3b) showed 9-fold lower anti-His antibody responses with the use of CoP1 liposomes (2.27 $\log_{10}$ luminescence (LUM)), confirming more stable Env attachment compared to Ni-NTA liposomes (3.24 $\log_{10}$ LUM) (Supplementary Fig. 3c; Supplementary Table 1). In congruence, in the current study, all Env-CoPoP immunized groups exhibited comparably low anti-His antibody titers ranging from 2.52 to 2.70 $\log_{10}$ median LUM values (Fig. 2d).

### Broadly reactive tier 2 neutralization elicited by HIV-1 Env-CoPoP liposomes

To analyze the neutralization breadth and potency after a three-dose Env-CoPoP regimen, rabbit sera from week 18 were tested against Env-pseudotyped viruses in the TZM-bl neutralization assay. The test panels included (i.) a clade C panel with tier 2 ($n = 9$) and tier 1B ($n = 1$) strains to study clade-specific responses induced by the immunogens used, (ii.) the global panel of tier 2 ($n = 12$) strains[52] to study cross-clade responses and assess breadth of the response. All sera were tested against an MLV Env-pseudotyped virus as negative control to confirm specificity of the neutralizing response.

Most animals in the three Env-CoPoP immunized groups exhibited serum neutralizing antibody responses, with the responses of the control animals staying below the limit of detection (serum $ID_{50} < 20$). The tier 1B strain ConS and the easy-to-neutralize tier 2 strain TV1.21 were neutralized efficiently by all Env-CoPoP groups with combined geometric mean titers (GMTs) of 704 (Env-CoP1), 1533 (Env-CoP2) and 366 (Env-CoP3) serum $ID_{50}$ (Fig. 3a; Supplementary Table 2). Though low to moderate in intensity and not observed in all animals, serum neutralization was observed against 18 of 20 tested difficult-to-neutralize tier 2 strains across the three groups, demonstrating that multi-clade immunity against diverse HIV-1 strains was induced by the Env-CoPoP liposomes. This response was not confined to strains similar to the three clade C immunogens used, but also spanned strains that are phylogenetically distant (Supplementary Fig. 4a and 4b). Of note, a clear difference in neutralizing activity was observed when CoP1 and CoP2 particles were compared with the CoP3 liposomes, with the latter inducing significantly lower serum neutralizing antibody titers (Fig. 3a, Fig. 3b, Supplementary Table 1). While vaccination with Env-CoP3 particles induced some neutralization, including cross-clade neutralization, not all animals responded. In contrast and albeit at low levels, all animals in the Env-CoP1 and Env-CoP2 groups exhibited a neutralizing antibody response that spanned at least four or more tier 2 HIV-1 strains. There was a wide range in the neutralization levels, which varied from 20-50 to over 500 serum $ID_{50}$ in some animals. To predict the Env epitopes targeted by the antibodies in the polyclonal sera of CoPoP immunized animals, we used neutralization fingerprints[53] of a panel of bNAbs against the tested tier 2 strains (Fig. 3c, Supplementary Table 3). These analyses revealed PGT151- and 8ANC195-like patterns in several animals, indicating the predominance of fusion peptide and gp120-gp41 interface antibodies (Fig. 3c). Of note, the prevalence of signatures resembling V1/V2 loop, V3 loop, CD4bs and MPER antibodies was higher in Env-CoP1 and Env-CoP2 immunized animals compared to the Env-CoP3 group (Fig. 3c), corroborating the higher neutralization breadth seen in the former groups (Fig. 3a).

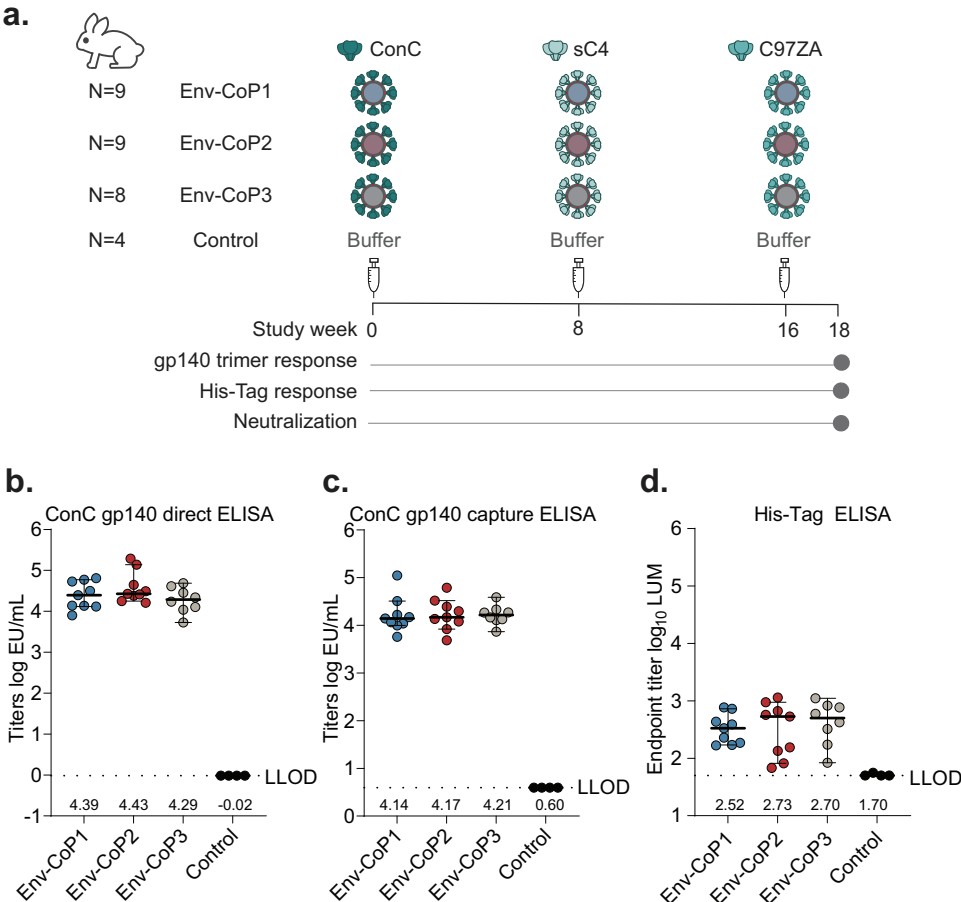

**Fig. 2 | In vivo immunogenicity of CoPoP HIV-1 Env particles in NZW rabbits.**
**a** Schematic of study design with details on immunization schedule of rabbits receiving the different Env-CoPoP immunogens adjuvanted with Adjuplex along with the control group receiving buffer. **b** Direct ELISA titers of immunized rabbits at week 18 against the ConC gp140 trimer to detect total binding antibodies. **c** Capture ELISA titers of immunized rabbits at week 18 against the ConC gp140 trimer to detect antibodies binding to conformational epitopes. **d** ELISA titers of

immunized rabbits at week 18 against the His-tag to assess off-target anti-His responses indicating Env detachment. Bars in **b**–**d** represent median with 95% confidence interval (CI) for individual rabbits within the Env-CoP1 ($N = 9$), Env-CoP2 ($N = 9$), Env-CoP3 ($N = 8$) and control ($N = 4$) groups; and the dotted lines represent the ELISA assay lower limit of detection (LLOD). Numbers at the bottom of the graphs **b**–**d** denote median values. Source data are provided as a Source Data file.

For the analysis of neutralization breadth and potency of the neutralizing antibody responses, we only included the 20 tested tier 2 viruses (also excluding the easy-to-neutralize clade C strain TV1.21) (Fig. 3b). The potency and breadth between the groups were assessed by quantifying the GMT and % breadth (Fig. 3b). These analyses revealed similar potency (GMT of 20 and 19 $ID_{50}$) and neutralization breadth (35% each) for the Env-CoP1 and Env-CoP2 groups, respectively (Fig. 3b; Supplementary Table 1). In comparison, the Env-CoP3 group exhibited lower GMT (13 $ID_{50}$) and neutralization breadth (13%; Fig. 3b; Supplementary Table 1). The GMT values were calculated by censoring and assigning a value below LLOD for data points with no detectable neutralization (see methods). Therefore, GMT values of 20, 19 and 13 for Env-CoP1, Env-CoP2 and Env-CoP3 respectively, though low due to censored values, provide a groupwise comparison of the serum neutralization against the 20 tested tier 2 strains (Supplementary Table 1).

Finally, we wanted to study the relationship between total binding antibody levels and neutralizing responses. To this end, a comparison of ELISA titers and neutralizing activity revealed limited correlation between binding antibody levels and serum $ID_{50}$ levels (Spearman r 0.4643; p value 0.0169) and no significant correlation between binding antibody levels and neutralization breadth (Spearman r 0.3305; p value 0.0992) (Supplementary Fig. 4c).

## Ad26 encoding HIV-1 Env can induce recall antibody responses in liposome primed rabbits

By presenting antigens in a different format, heterologous vaccination regimens can induce broader diversity in the immune responses against viruses as also evidenced recently during the SARS-CoV-2 pandemic[54,55]. Therefore, we generated Ad26 vectors encoding membrane-bound RnS HIV-1 ConC, sC4 and C97ZA clade C Envs. These transgenes included a consensus clade C transmembrane domain and C-terminus that was truncated after position 712 to create high expressing, membrane-anchored HIV-1 Env trimers[56]. To mitigate the need for sufficient furin expression in vivo, cleavage-independent designs were created with a 2x[G4S] linker[56,57] (Supplementary Fig. 5a). The ConC, sC4 and C97ZA Env encoding Ad26 vector particles were mixed in a 1:1:1 ratio to generate a trivalent cocktail termed Ad26.RnS.Env. The antigenic profile of Ad26.RnS.Env was measured after transduction of A549 cells and surface staining with a panel of bNAbs and non-bNAbs was assessed by flow cytometry (gating strategy in Supplementary Fig. 5b), demonstrating binding to bNAbs and minimal binding to non-bNAbs (Fig. 4a).

To test the immunogenicity of a heterologous vaccine platform using the Ad26.RnS.Env cocktail, the Env-CoP2 group along with the control group were selected. Rabbits in the Env-CoP2 group were dosed via the i.m. route with a total of $5 \times 10^9$ viral particles (vp) at week

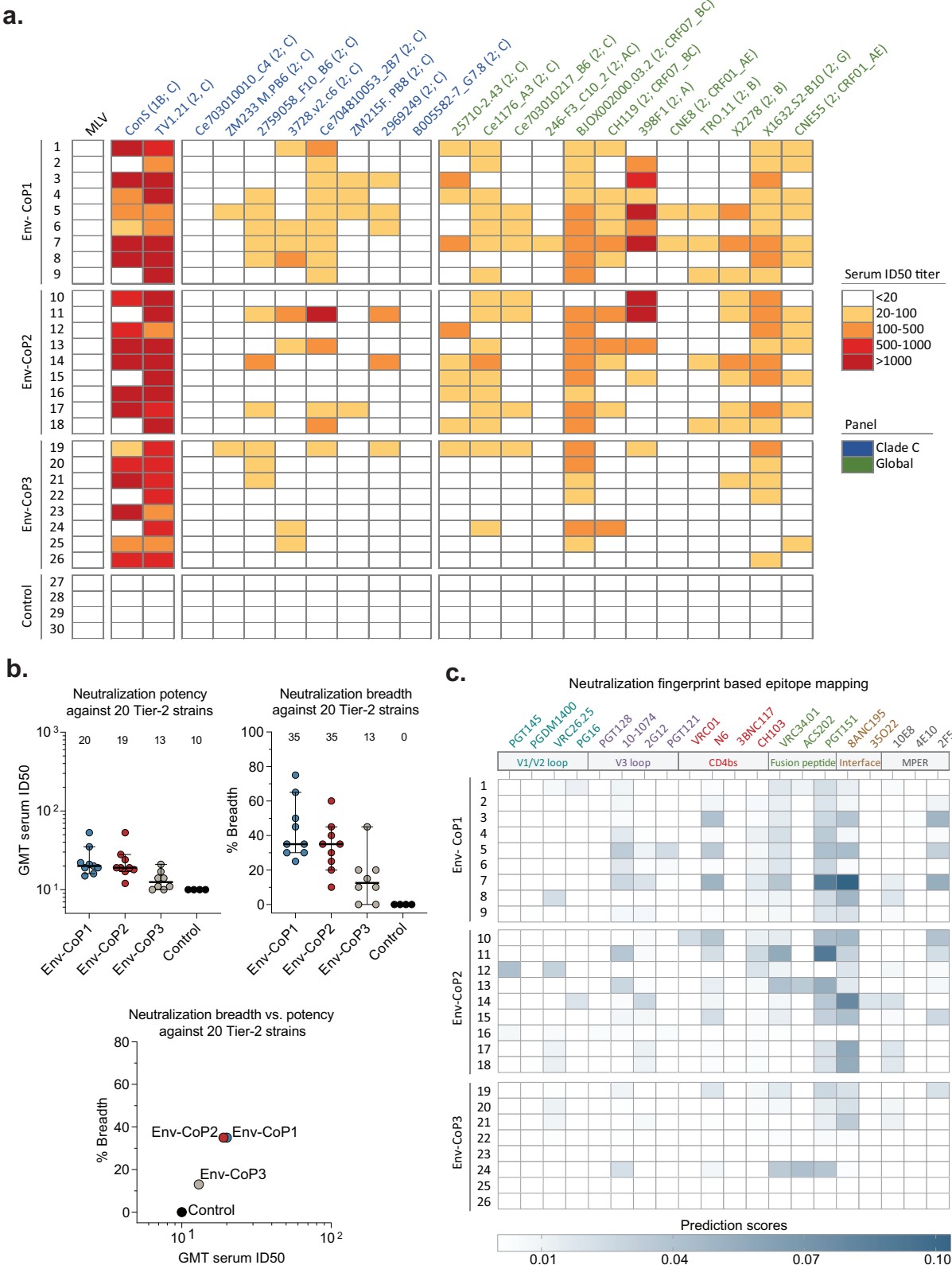

34 and control animals were administered with buffer (Fig. 4b). Overall levels of antibodies binding to HIV-1 Env were assessed using a direct coat ELISA to test binding to ConC gp140 trimers at the time of Ad26 dose (week 34) and 2 weeks post dosing at week 36.

All animals immunized with Ad26.RnS.Env showed a recall response of total antibodies binding to the HIV-1 Env trimer, which was comparable in magnitude to that observed at peak immunogenicity (week 18) after the particle regimen (median titers of 4.42 and 4.40 log EU/ml at week 18 and week 36 respectively; Fig. 4c; Supplementary Table 1). Similarly, antibody responses against conformational epitopes, determined using a capture ELISA, showed similar levels at week 36 (4.47 log EU/ml) compared with the Env-CoPoP liposome peak at week 18 (4.16 log EU/ml; Fig. 4d; Supplementary Table 1).

**Fig. 3 | Serum neutralizing activity following HIV-1 Env-CoPoP particle immunization. a** Color gradient map depicting the serum neutralizing activity of sera from week 18 blood draws against Env pseudotyped viruses with Murine Leukemia Virus (MLV) as negative control, a clade C panel and the global panel, measured in the TZM.bl neutralization assay. The tier and clade of each pseudovirus Env strain is shown in brackets. **b** Top left: scatter plot showing geometric mean titer (GMT) neutralization titers per rabbit within the Env-CoP1 ($N = 9$), Env-CoP2 ($N = 9$), Env-CoP3 ($N = 8$) and control ($N = 4$) groups, calculated as the GMT of serum neutralization against the n = 20 difficult-to-neutralize tier 2 HIV-1 strains used in panels. Top right: scatter plot showing neutralization breadth per rabbit, calculated as the % of strains neutralized out of the n = 20 tested tier 2 strains. Bottom: dot plot

showing GMT neutralization vs. median % breadth for all groups in the study. Values below the neutralization assay lower limit of detection (LLOD) were set to 1/2 the LLOD of 10 serum $ID_{50}$ for plotting and analysis. Bars in left/middle graphs in **b** represent median with 95% confidence interval (CI) and the dotted lines represent the ELISA assay LLOD and numbers at the top of the graphs denote median values. **c** Neutralization fingerprint-based epitope prediction against major HIV-1 Env bNAb binding sites for week 18 sera. Numbers next to group allocation in **a** and **c** denote animal ID. Source data are provided as a Source Data file. bNAb: broadly neutralizing antibody; CD4bs: CD4 binding site; MPER: membrane proximal external region.

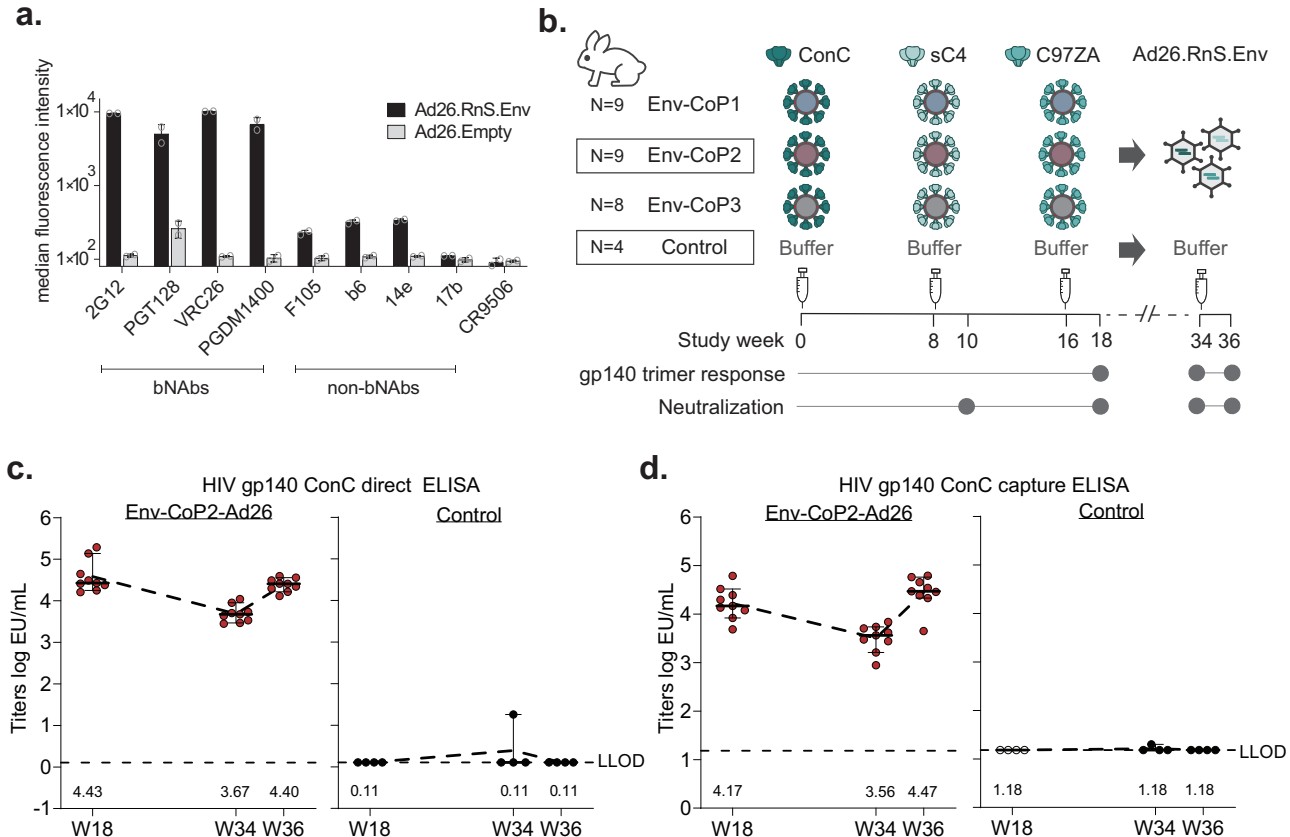

**Fig. 4 | Ad26 encoding HIV-1 Env can boost particle prime induced antibody responses. a** Antigenic binding profile of a trivalent mixture of Ad26.RnS.Env, encoding membrane-expressed clade C ConC, sC4, and C97ZA. Binding to a panel of broadly neutralizing antibodies (bNAbs) (2G12, PGT128, VRC26, and PGDM1400) and non-bNAbs (F105, b6, 14e, and 17b) was assessed by flow cytometry. CR9506 antibody was used as an isotype control. Bars represent average ± standard deviation (SD) of $n = 2$ technical replicates (open circles). **b** Schematic of Ad26 boost phase; the Env-CoP2 and control groups (highlighted boxes) were immunized with Ad26.RnS.Env or buffer respectively, at week 34 and blood sampling for

immunogenicity was done two weeks later at week 36. **c** Direct ELISA titers of immunized animals against the ConC gp140 trimer to detect all binding antibodies in the Ad26 boost phase at weeks 34 and 36 in comparison to week 18. **d** Capture ELISA titers against the ConC gp140 trimer to detect antibodies binding to conformational epitopes in the Ad26 vs. particle immunization phases. Bars in **c**, **d** represent median with 95% confidence interval (CI) for individual rabbits within the Env-CoP2-Ad26 (N = 9) and control (N = 4) groups, dotted lines represent the ELISA assay LLOD and numbers at the bottom of the graphs denote median values. Source data are provided as a Source Data file.

## Ad26 HIV-1 Env re-triggers tier 2 clade C neutralizing antibody responses

Next, we assessed the ability of the Ad26-encoded clade C Env immunogens to induce recall of the tier 2 neutralizing antibody responses seen after the CoPoP particle immunization phase. To this end, we analyzed the kinetics of serum neutralization of the Env-CoP2 and control groups starting at week 10, which is two weeks after the second CoPoP-Env immunization and followed it up to week 36, which is two weeks post Ad26.RnS.Env dosing (Fig. 5). This longitudinal analysis was performed by testing the sera against Env-pseudotyped viruses expressing the clade C Envs and the MLV Env (Fig. 5).

The serum neutralization induced by the Env-CoPoP immunizations declined over time between week 18 and week 34 (Fig. 5). The Ad26 boost recalled serum neutralizing antibody responses against clade C virus strains (week 36) to a similar magnitude as that observed after three particle immunizations (week 18) (Fig. 5). Nearly all animals responded to the tier 1B ConS strain and the easy-to-neutralize tier 2 strain TV1.21, and four of eight tier 2 clade C strains were neutralized at week 36, following the Ad26 boost (Fig. 5).

The breadth and potency against the clade C panel (assessing only the 8 difficult-to-neutralize tier 2 strains) at week 36, following the single Ad26.RnS.Env dose was similar to that at week 18 after particle

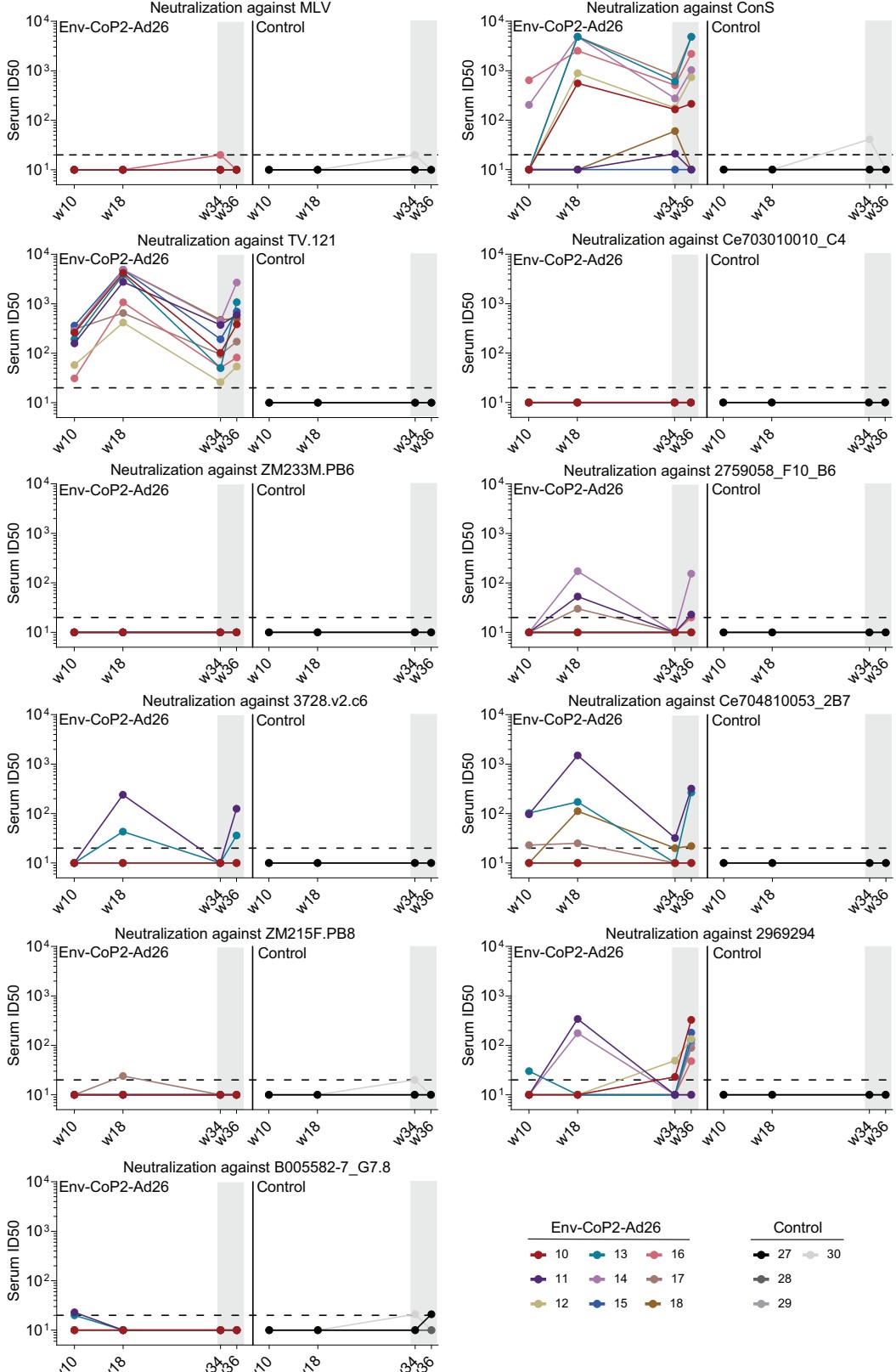

**Fig. 5 | Ad26 HIV-1 Env boosts tier 2 Clade C neutralizing antibody responses.**
Longitudinal kinetics of serum neutralizing activity of Env-CoP2 ($N = 9$) and control ($N = 4$) groups through the course of the three-dose particle immunization phase (weeks 10, 18) into the Ad26 boost phase (week 34, 36) against the clade C pseudovirus panel of $n = 10$ strains and Murine Leukemia Virus (MLV) Env pseudotype as negative control. Each color denotes one animal and numbers represent animal ID. Gray regions indicate the Ad26 boost phase. Dotted lines represent the neutralization assay lower limit of detection (LLOD) of 20 Serum $ID_{50}$ and values below the LLOD were set to 1/2 the LLOD of 10 serum $ID_{50}$ for plotting and analysis. Source data are provided as a Source Data file.

immunization (Supplementary Fig. 6). At both week 36 and week 18, the observed neutralization breadth and GMT neutralization were the same, that is 13% and 14 $ID_{50}$, respectively (Supplementary Fig. 6).

## Discussion

An effective HIV-1 vaccine should be able to build an immune barrier to block infection and provide sterilizing immunity. The complexities of the HIV-1 Env like high antigenic diversity, inherent protein instability, and epitope masking by glycosylation[4,10] likely make a multi-component vaccine approach crucial to achieving a sufficiently potent and broad immunity against this virus. In this study, we describe an HIV-1 vaccine candidate consisting of CoPoP liposomal particles displaying one of three different RnS clade C Env immunogens[22] which, when administered 8 weeks apart in sequential order, induced high titer binding antibody responses. Importantly, tier 2 neutralizing antibody responses could be induced with the use of these clade C immunogens against strains spanning multiple clades of HIV-1. Additionally, the antibody binding and neutralization responses could be recalled with a heterologous vaccine platform using genetically-encoded Env antigens as a boost.

Delivery of antigens on CoPoP liposomes has been demonstrated to enhance the immune response by several orders of magnitude compared with soluble immunogens as a result of the particulate presentation[45,58]. Additionally, CoPoP liposomes provide versatility in terms of adjuvant incorporation[45] and were proven to be stable and functional following lyophilization and reconstitution[59]. We have generated CoPoP liposomes presenting a dense array of stably attached RnS HIV-1 Env trimers in a prefusion conformation, which could be concluded based on a high quality Env antigenicity profile. Binding of a broad panel of antibodies against major bNAb epitopes on HIV-1 Env, including the V1/V2 and V3 region, CD4bs, fusion peptide, and gp120-gp41 interface, in the absence of binding to non-bNAbs was confirmed. However, sC4 and C97ZA showed reduced binding to CD4bs antibodies. These variants contain a disulfide bridge which has been described previously to impact binding of the CD4bs class of antibodies[60]. The stabilized HIV-1 Env antigens exhibited high thermostability, a characteristic which has been linked to induction of an improved neutralizing antibody response[61]. Using recombinant HIV-1 Env protein antigens provides the benefit of purifying correctly folded closed Env trimers, which is not possible for vector-expressed membrane-bound HIV-1 Env immunogens in host cells in vivo. This is illustrated in this study by the absence of binding of non-bNAbs to soluble RnS HIV-1 Env proteins, while residual binding of non-bNAbs to Ad26 expressed RnS HIV-1 Env could be detected.

In this study, all three CoPoP formulations induced high titers of clade C gp140 Env trimer binding antibodies in serum of immunized rabbits. However, the high levels of binding antibody titers did not directly translate into broad and potent neutralizing activity for all groups, with Env-CoP1/Env-CoP2 immunized animals showing superior serum neutralizing activity compared to the Env-CoP3 group. This outcome indicates that differences in particle composition did not affect the generation of binding antibodies but did result in differential bNAb responses, possibly due to alternate B cell maturation pathways being triggered[62]. In this regard, the CoP3 particles differ in their lipid composition by utilizing DOPC instead of the DPPC phospholipid of CoP1/CoP2, due to the tendency for QS-21 to induce aggregation of DPPC-based liposomes[58]. Based on other studies, differences in lipid composition can affect membrane rigidity and influence immunogenicity which might explain these observations[27,63]. The lack of a clear increase in diameter of CoP3 particles following Env decoration could indicate that the coupling procedure impacts CoP3 liposomes differently compared with CoP1/CoP2 particles. Moreover, we observed lower PGT145 binding to Env-CoP3 liposomes, suggesting compromised Env integrity in the CoP3 formulation compared with other particles. The combined in vitro observations and suboptimal in vivo

performance of the CoP3 particle disfavor this formulation, although the impact of individual liposome components warrants further evaluation.

The average breadth coverage of neutralizing responses elicited by the HIV-1 Env-decorated CoPoP particles was up to 35% against a multiclade panel of n = 20 tier 2 strains, demonstrating promise when compared to the ability of other vaccine candidates to induce poly-clonal responses[28–30,64–66] or monoclonal antibodies[67–70] in animal models. Neutralization fingerprint-based epitope mapping of sera revealed that the CoPoP regimen could generate neutralizing antibodies that are predicted to target various conserved bNAb binding regions of the Env including the V1/V2 loop, V3 loop, CD4bs, fusion peptide, gp120-gp41 interface and MPER[43]. The predicted reactivity of CoPoP immunized sera was highest against the fusion peptide and gp120-gp41 interface regions of the Env. Even though our immunogens only contain few of the N-terminal residues of the MPER, the pattern of predicted neutralization signatures resembling MPER antibodies 2F5 and 10E8 might stem from the large Env binding footprint of bNAbs that extends to upstream, more membrane-distal residues, for 2F5[71] and even gp120 for 10E8[72]. Of note, despite the sole use of clade C immunogens in this study, neutralization against non-clade C tier 2 strains could be achieved in the majority of rabbits, underlining the potential of RnS Env immunogens[22] in efficiently presenting shared neutralizing antibody epitopes. Germline targeting immunogens were recently shown to elicit rare bNAb lineages in humans[17]. In an alternative strategy, here we present a consensus-based approach utilizing RnS prefusion closed trimers, which provide the epitope availability needed for triggering and expanding bNAbs and inducing serum HIV-1 neutralizing activity in rabbits. Further investigation into the specific B cell lineages induced by the CoPoP vaccine regimen would shed light on the bNAb development pathways triggered.

Besides neutralizing breadth, sufficiently high levels of serum neutralization would also likely be key to an effective HIV-1 vaccine[10]. Weak tier 2 neutralization elicited by sequential immunization with VLP-based immunogens did not protect against high-dose SHIV challenge in NHPs[68]. Though several rabbits in this study displayed serum titers between 100-500 $ID_{50}$ and some even above 500 $ID_{50}$, the group GMT against the 20 tier 2 strains was low due to heterogeneity in response rates, indicating need for improvement. Moreover, the serum bNAb responses elicited by the HIV-1 Env-complexed CoPoP liposome platform contracted over time. This points to the need for a booster immunization to maintain long-term protective levels. Previous studies have shown the benefits of heterologous immunization regimens in increasing the breadth of antibody responses to viral antigens[54,55]. With the aim to generate immunity with increased potency and breadth, we tested an HIV-1 vaccine approach using a heterologous regimen consisting of protein and genetically encoded immunogens. We selected the CoP2 group to test the heterologous boost due to its superiority in Env loading efficiency and higher Env decoration over CoP1, despite being similar in a head-to-head comparison of elicited neutralization breadth and mean neutralization titers. The Ad26.RnS.Env vector-based vaccination could efficiently recall both binding antibody titers as well as the tier 2 serum neutralization. Although we did not observe additional increase in breadth or magnitude of the neutralizing antibody response after Ad26 boost immunization of Env-CoPoP immunized rabbits, the breadth of neutralization after recall was not limited to the easy-to-neutralize strains TV1.21 or ConS. Of note, we utilized a trivalent cocktail for the boost, which may not be as efficient as single immunogens in eliciting recall responses[29]. Additional aspects like interval optimization[73] as well as gradient antigen dosing[74] in the vaccine regimen might further improve breadth and potency.

Despite the recent insights from the VRC01 AMP trials that provided proof-of-concept that bNAbs could be protective against sensitive strains[75], a clear understanding of NAb correlates of protection for

HIV-1 vaccines in humans remains an open question in the field. Thus, it remains to be seen how the humoral immune response induced by the CoPoP regimen assessed here translates into humans and whether it can provide protection, particularly in the context of eliciting rare bNAb B cell lineages.

A limitation of this study is the lack of direct evidence of induction of T cell immunity following HIV-1 Env-CoPoP immunization due to limited tools to screen cellular immunity in rabbits. Particularly, T follicular helper (Tfh) cells were shown to aid B cell maturation in the germinal centers by facilitating somatic hypermutation and affinity maturation that drives bNAb development from rare B cell lineages[62,76]. CoPoP liposomes presenting other viral antigens have been shown to induce these CD4+ Tfh cells in mice[45] and the Adjuplex used as adjuvant here is also known to aid T cell immunity[46,47]. Thus, further work is needed to understand T cell responses in the context of the HIV-1 Env-CoPoP vaccine regimen.

In this study, we could show that sequential immunization with three stabilized clade C HIV-1 RnS Envs presented on CoPoP liposomes has the potential to induce cross-clade tier 2 neutralizing antibodies. Moreover, the CoPoP liposome-based protein antigen delivery can be combined with other HIV-1 vaccine strategies in a heterologous vaccine regimen to induce optimal bNAb responses. Such an approach provides the potential to induce broadly reactive and neutralizing anti-HIV-1 antibody responses as part of an effective HIV-1 vaccine.

# Methods

## Design of HIV-1 Env antigen sequences

Soluble HIV-1 Env immunogens of ConC, sC4, C97ZA, and BG505 were designed as previously described, with some modifications[22] (Supplementary Fig. 1). All immunogens had an eight-histidine residue C-terminal tag (His-tag) for liposome coupling. Membrane-anchored HIV-1 Env designs of ConC, sC4 and C97ZA were based on soluble designs as described above, with some modifications (Supplementary Fig. 5). In brief, designs contained a consensus C-based transmembrane domain and truncated C-terminus after position 712 and a 2x[G4S] linker replacing the furin cleavage site (positions 508-511). All sequences had SOS substitutions removed as well as a PNGS removed that is less common (<20%) for clade C strains. Finally, three additional stabilizing mutations, N302M, R304V and T320L were introduced in all designs[77,78].

## Expression and purification of soluble HIV-1 Env proteins

HIV-1 Env encoding constructs were synthesized and codon-optimized at GenScript (Piscataway, NJ 08854) and cloned into pcDNA2004. Expression and purification of recombinant proteins was carried out by U-Protein Express B.V. (now ImmunoPrecise Antibodies, The Netherlands). Briefly, HEK293E-253 cells were transiently transfected with HIV-1 Env, furin, and carrier DNA in a 1:1:8 ratio and harvested six days post transfection. Culture supernatant was harvested and clarified by centrifugation. Recombinant HIV-1 Envs were purified via a three-step purification process, applying a CAP259 VRC26.09 column for initial affinity purification followed by a Superdex200 16/600 gel filtration column. Fractions containing trimeric recombinant protein were pooled and subjected to a second, negative-selection affinity purification step using a 14e column. The flow-through containing correctly folded recombinant HIV-1 Env trimers was collected and sterilized using a 0.22 μm syringe filter.

## BioLayer Interferometry (Octet)

Binding of HIV-1 Envs to immobilized antibodies was measured with an Octet HTX instrument (Sartorius GmbH, Göttingen, Germany). Monoclonal antibodies used for evaluation were purchased from Polymun Scientific (2G12, 447-52D) or produced as human IgG using published sequences of the variable domains for the following antibodies with varying specificities: V1/V2-apex/trimer specific: PGT145[79],

PGDM1400[80], VRC26[81], V3-base: PGT128[79], 2G12[82], V3 crown: 14e[83], 447-52D[84], CD4-binding site: VRC01[85], 3BNC60[86], b6[87], F105[88], CD4-induced: 17b[89], fusion peptide: VRC34[90], PGT151[91], gp120-gp41 interface: 35O22[92]. All steps were performed in 1× kinetics buffer (FortéBio) in 384-well tilted-bottom plates (FortéBio) at 30 °C, shaking speed 1000 rpm. HIV-1 Env antibodies were immobilized on anti-hIgG (AHC) sensors (FortéBio) for 600 s at a concentration of 10 μg/mL, followed by a baseline of 300 s. Association of HIV-1 Env protein was measured for 300 s at a concentration of 10 μg/mL, followed by a dissociation step of 300 s. Data analysis was performed by calculating the R-equilibrium at 300 s association using FortéBio Data Analysis 12 software (FortéBio) from three technical replicates of a single experiment.

## Nano Differential Scanning Fluorimetry (nanoDSF)

Protein thermostability was evaluated by nanoDSF using an Uncle instrument (Unchained Labs). Proteins at a concentration of 0.5 mg/mL were subjected to a 20–90 °C thermal ramp of 1 °C per minute. The melting temperature ($Tm_{50}$) was determined based on the barycentric mean of intrinsic tryptophan fluorescence emission spectra collected from the thermal ramp and was calculated by Uncle Analysis software v5.03 (Unchained Labs) from a single experiment.

## HIV-1 Env liposome preparation and characterization

Cobalt porphyrin-phospholipid (CoPoP) was produced as previously described[93]. The other lipids used for CoPoP liposomes were: 1,2-Dioleoyl-sn-glycero-3-phosphocholine (DOPC, Corden, catalog number LP-R4-070), 1,2-Dipalmitoyl-sn-glycero-3-phosphocholine (DPPC, Corden, catalog number LP-R4-057), 1,2-distearoyl-sn-glycero-3-phosphocholine (DSPC, Corden, catalog number LP-R4-076), cholesterol (PhytoChol, Wilshire Technologies, CAS 57-88-5), and Monophosphoryl Hexa-acyl Lipid A, 3-Deacyl (Synthetic) (PHAD-3D6A, Avanti, catalog number 699855, abbreviated as PHAD herein). QS-21 was obtained from Desert King. Components used for Ni-NTA liposomes were: DSPC (Avanti, catalog number 850365), 1-palmitoyl-2-cholesterylcarbonoyl-sn-glycero-3-phosphocholine (PChcPC, Avanti, catalog number 880345) and 1,2-dioleoyl-sn-glycero-3-[(N-(5-amino-1-carboxypentyl)iminodiacetic acid)succinyl] (Ni-NTA, Avanti, catalog number 790404). CoPoP liposomes were prepared by ethanol injection followed by nitrogen-pressurized lipid extrusion[45]. In brief, lipids were weighed, dissolved in 1 mL of absolute ethanol, briefly sonicated in a water bath sonicator for less than 10 s to break up larger particles, and then heated at 55 °C for 10 min until all lipids were fully dissolved. 4 mL of PBS (pre-heated to 55 °C) was rapidly injected, and the mixture was briefly sonicated, incubated at 55 °C for another 10 min, and then briefly sonicated again prior to extrusion. The liposome mixture was passed through stacked 200 nm, 100 nm, and 80 nm polycarbonate membranes (Whatman) 10 times in a lipid extruder (Lipex, Northern Lipids) at 55 °C with an observed pressure of 200–300 PSI. Following extrusion, ethanol was removed by dialysis at 4 °C against 1 L of PBS twice. Liposomes were finally passed through a 0.2 μm sterile filter and stored at 4 °C. The liposome size and polydispersity index were determined by dynamic light scattering (DLS) with an Uncle instrument (Unchained Labs) and reported as average ±SD of three replicate measurements. Formulation CoP1 consisted of DPPC, cholesterol (Chol), and CoPoP [DPPC:Chol:CoPoP at mass ratio 4:2:1]. CoP2 additionally incorporated the adjuvant 3D6A-PHAD and contained increased CoPoP content [DPPC:Chol:PHAD:CoPoP at mass ratio 4:2:1:3.7]. CoP3 incorporated both 3D6A-PHAD as well as QS-21 adjuvant. To ensure compatibility with QS-21, CoP3 was created with DOPC instead of DPPC [DOPC:Chol:PHAD:QS-21:CoPoP at mass ratio 4:2:1:1:3.6]. Ni-NTA liposomes were prepared by Avanti Polar Lipids and consisted of DSPC, pChcPC and Ni-NTA [DSPC:PChcPC:Ni-NTA at mass ratio 11.4:7.9:1]. Aiming at high Env trimer density and an efficient coupling reaction, HIV-1 Env liposomes were prepared by incubating

HIV-1 Env trimer and liposomes at a mass ratio of [Env:liposome] [2:1] for CoP1, [1.35:1] for CoP2, [1.46:1] for CoP3, and [4:1] for Ni-NTA. The HIV-1 Env-CoPoP mixtures were incubated for 20 minutes at 50 °C to increase membrane fluidity and thereby improve coupling efficiency and were then incubated overnight at room temperature (RT). To determine coupling efficiency and calculate HIV-1 Env trimer density, a fraction of the mixture was incubated with Ni-sepharose beads (GE Healthcare) to remove unbound Env. The resulting HIV-1 Env-liposome bound fraction was subjected to reduced SDS-PAGE and stained with Coomassie, and gp120 bands were quantified using a Li-Cor Odyssey instrument. HIV-1 Env trimer decoration was calculated per 100 nm diameter liposome, by correcting the calculated trimer density for the obtained coupling efficiency and was reported as average Env trimer ±SD per liposome type. For in vivo application, the remaining mixture was subjected to purification by size exclusion chromatography (SEC) using a Sepax SRT-10C 300 column on an Akta Avant (Cytiva). The HIV-1 Env-liposome peak was collected in Tris buffer (25 mM Tris, 150 mM NaCl pH7.4) and sterile-filtered through a 0.2-micron SCFA filter (Corning). A dilution series of purified HIV-1 Env liposomes was subjected to reduced SDS-PAGE and stained with Coomassie, and gp120 bands were quantified to determine the HIV-1 Env concentration of the purified product. The liposome size and polydispersity index were determined by three replicate DLS measurements with an Uncle instrument (Unchained Labs) and reported as average ±SD per liposome type. A representative dataset was selected from four independent experiments.

### Cryo Electron Microscopy

Vitrification of purified HIV-1 Env liposomes was done using a Vitrobot Mark IV (Thermo Fisher Scientific). 3 µL sC4-CoP1, sC4-CoP2, or sC4-CoP3 sample was applied onto freshly glow discharged Quantifoil R2/2 copper grids, blotted for 3 seconds at 22 °C and 100% relative humidity, and plunge frozen in liquid ethane. Micrographs were collected on a Titan Krios operated at 300 kV, equipped with a Falcon 3EC detector (Thermo Fisher Scientific) at a nominal magnification of 19500 x (4.4 angstrom per pixel) at The Netherlands Center for Electron Nanoscopy (NeCEN). Two representative images per grid were selected from a single experiment.

### AlphaLISA

AlphaLISA is a bead-based proximity assay in which singlet oxygen molecules generated by high energy irradiation of donor beads transfers to acceptor beads, which are within a distance of ~200 nm. A cascading series of chemical reactions results in a chemiluminescent signal[94]. Antibodies PGT128, PGT145 and F105 were equipped C-terminally with either a biotin label or with a V5 label using sortase-directed coupling and were combined with streptavidin (SA) donor beads and anti-V5 acceptor beads. AlphaLISA assay was carried out in PBS + 0.05% Tween-20 + 0.5 mg/mL bovine serum albumin (BSA) buffer, containing a dilution series of HIV-1 Env (soluble or liposome-displayed), together with 3 nM biotin-labeled antibody, 3 nM V5-labeled antibody, and 5 µg/mL SA donor bead (PerkinElmer) and 5 µg/mL anti-V5 acceptor bead (PerkinElmer). For His-tag detection, the HIV-1 Env dilution series was combined with 3 nM biotin-labeled antibody and 5 µg/mL SA donor bead and 5 µg/mL anti-His acceptor bead (PerkinElmer). After 2 hr incubation at RT the signal was measured on a PerkinElmer Ensight plate reader and collected with PerkinElmer Kaleido software version 3.0.3067.117. Data analysis was performed by determining the area under the curve obtained for AlphaLISA counts of the 6-step HIV-1 Env dilution series, applying a threshold for AlphaLISA counts obtained with an aspecific biotin-labeled anti-RSV antibody CR9506 (Janssen, in-house production) in combination with PGT128-V5 (SA donor and anti-V5 acceptor combination) or with biotinylated CR9506 alone (SA donor and anti-His acceptor combination) (GraphPad Prism 9.0.0 software). A representative dataset was selected from three

independent experiments. To assess serum stress resilience, HIV-1 Env liposomes were incubated overnight at 37 °C in a final concentration of 0, 20, or 50% rabbit serum (Innovative Research) before being subjected to AlphaLISA assay. AlphaLISA signal of 2-3 replicate measurements from a single experiment obtained with 0.6 nM HIV-1 Env dilution were plotted after subtraction of blank (0 nM Env).

### Production and characterization of Ad26 vectors expressing HIV-1 membrane Env immunogens

Replication incompetent, E1/E3 deleted Ad26 vectors, where the E1 gene was replaced by a transgene cassette were used. The transgene cassette consisted of a TetO containing human CMV promotor followed by the transgene, encoding the membrane-bound versions of ConC, sC4 or C97ZA Env genes and an SV40 PolyA. The vectors were propagated in PERC6.TetR cells and purified using a two-step CsCl centrifugation procedure. The Ad26 products were formulated in VGII buffer (10 mM Tris (pH 7.4), 1 mM MgCl2, 75 mM NaCl, 5% sucrose, 0.02% PS-80, 0.1 mM EDTA, 10 mM Histidine, 0.5% ETOH) and stored at -80 °C. Quantification of virus particle (vp) titers was done by measurement of optical density at 260 nm and infectivity was assessed by quantitative potency assay (QPA) using PER.C6 TetR cells[95]. Virus integrity and identity were confirmed by PCR on adenovirus and transgene genes and sequencing of the entire expression cassette. Transgene expression was confirmed by western blotting. The chromogenic limulus amebocyte lysate (LAL) assay was used to check for endotoxin levels and bioburden was assessed by membrane filtration testing.

To characterize HIV-1 Env expression, A549 cells were transduced with a 1:1:1 mixture of three Ad26 vectors (Ad26.RnS.Env) encoding HIV-1 Env transgenes ConC, sC4, and C97ZA at a total of 25.000 vp/cell for 48 h. As a negative control, A549 cells were transduced with 25.000 vp/cell Ad26.Empty vector, not encoding a transgene. For HIV-1 Env surface staining, harvested cells were washed with PBS and stained with LIVE/DEAD™ Fixable Violet Dead Cell Stain Kit (Invitrogen). After washing with FACS buffer (PBS with 0.5% BSA), cells were incubated with anti-gp120 antibodies 2G12 (5 µg/mL), PGT128 (0.2 µg/mL), VRC26 (1 µg/mL), PGDM1400 (2 µg/mL), F105 (5 µg/mL), b6 (2 µg/mL), 14e (5 µg/mL), and 17b (10 µg/mL) for 30 min. RSV antibody CR9506 was included as an isotype control at 1 µg/mL. Cells were washed twice with FACS buffer and stained with 1 µg/mL goat anti-human IgG Alexa Fluor 647 (Invitrogen) secondary antibody for 30 min. Cells were washed twice with FACS buffer and fixed with 1× BD CellFIX (BD Biosciences) for 15 min, washed and resuspended in FACS buffer and subjected to flow cytometry using a FACS Canto instrument (BD Biosciences). Data was collected with FACSDiva software v.9.0 (BD Biosciences) and analyzed with FlowJo_v10.7.1 Software (Becton, Dickinson and Company). Data was plotted as average median fluorescence intensity ±SD of the A549 single, live cell population of two replicate stainings, including a minimum of 23,000 gated cells. A representative dataset was selected from four independent experiments.

### Animals

16-18 weeks old (at first immunization) female New Zealand White rabbits were housed in an ARV BSL1 animal facility with 12 hourly light/day cycle and temperature controlled between 61-72 °F at Labcorp/Covance Laboratories, Inc. Only animals that demonstrated good health after physical examination were included in the study and study performed in compliance with the U.S. Department of Agriculture's (USDA) Animal Welfare Act (9 CFR Parts 1, 2, and 3); the Guide for the Care and Use of Laboratory Animals (Institute of Laboratory Animal Resources, National Academy Press, Washington, D.C., 2011); and the National Institutes of Health, Office of Laboratory Animal Welfare. Animal work was approved by the Labcorp Early Development Laboratories, Denver Site Institutional Animal Care and Use Committee (IACUC) review board.

## Immunizations and blood sampling

The dose formulations were prepared immediately prior to dosing and maintained on wet ice until the preparation of dosing syringes. In the CoPoP immunization phase, animals were administered 30 µg ConC, sC4 and C97ZA Env in the liposomal formulation of CoP1, CoP2 or CoP3 at week 0, week 8 and week 16 respectively (Fig. 2a). The Env liposomes were administered in Tris buffer (25 mM Tris, 150 mM NaCl pH7.5) adjuvanted with 20% v/v Adjuplex™ (Emperion LLC). In the Ad26 boost phase, animals from Env-CoP2 were administered with 5×10⁹ vp of Ad26.RnS.Env at week 34 of the study (Fig. 4b). The trivalent Ad26.Rns.Env was prepared by mixing equal amounts of Ad26.ConC, Ad26.sC4 and Ad26.C97ZA in VGII buffer (10 mM Tris pH 7.4, 1 mM MgCl2, 75 mM NaCl, 5% sucrose, 0.02% PS-80, 0.1 mM EDTA, 10 mM Histidine, 0.5% ETOH). All immunizations were performed via intramuscular administration in the right and left hind quadriceps at a dosing volume of 200 µl for each injection. The control group 4 animals were administered Tris buffer (25 mM Tris, 150 mM NaCl, pH7.5) at each dosing time point. Blood sampling was performed at weeks 0, 10, 18, 34, and 36 in aseptic conditions with a target volume of 7 mL, processed to serum and stored at -80 °C. Blood sampling on the day of vaccine dosing was performed before the immunization.

## Evaluation of HIV-1 Env specific antibody responses by direct coat ConC ELISA

A direct coat ELISA was performed to assess the binding HIV-1 Env specific antibody titers in rabbit sera. To this end, 0.3125 µg/ml biotinylated HIV gp140 ConC_SOSIP (Janssen) was coated onto 96 well plates in 1x PBS pH7.4 (Gibco) at 4 °C overnight. The plates were washed with PBST wash buffer (1x PBS with 0.05% Tween 20 (Millipore)) and blocked with blocking buffer (1x PBS with 2% BSA (Sigma) and 0.05% Tween 20 (Millipore)) for 1 h at RT. Rabbit sera, ConC standard serum, positive controls PGT145 and 447-52D mAbs (Janssen and Polymun respectively), and CR9056 mAb[96] (negative control; Janssen) were diluted in blocking buffer and 4-fold dilution series starting at 1:50 was performed for the sera and 5-fold dilution series starting at 40 µg/ml for the mAbs. The block buffer was removed from the plates and the sera or mAbs were incubated with the coated Ag for 1 h at RT. Next, plates were washed in wash buffer and incubated with 1:5,000 Goat anti-rabbit IgG HRP (Jackson ImmunoResearch) for serum samples and 1:15,000 Mouse anti-human IgG HRP (Jackson ImmunoResearch) for the mAbs for 1 h at RT. The plates were washed in wash buffer and incubated with LumiGLO Substrate A and B mix (Sera-care) for 30 min at RT in the dark. Luminescence was measured using the BioTek Synergy Neo plate reader. A nonlinear regression curve fit with 4-logistic parameters based on the average of duplicates linked to the dilution on the original scale was used to generate response curves. The relative potency (REP) to the reference standard was calculated by parallel line analysis with the Gen5™ Data Analysis Software which uses a constrained model for maximum response parameter to generate REP values which are the normalized by the EU/ml of the reference standard sera. A log10 transformation was done to plot the data on a linear scale. For calculating the lower limit of detection (LLOD) value, the luminescence values at the 1:50 start dilution of all Tris-buffer control group sera were compiled and the 95-percentile value was calculated at 0.26 of luminescence. All samples were measured in duplicate.

## Evaluation of HIV-1 Env conformational antibody responses by capture coat ConC ELISA

A capture coat ELISA was performed to assess titers of antibodies binding conformational epitopes on HIV-1 Env from the rabbit sera. To this end, 0.625 µg/ml Streptavidin (Thermo Fisher) was coated onto 96 well plates (Perkin Elmer) in 1× PBS pH7.4 (Gibco) at 4 °C overnight. The plates were washed with PBST wash buffer (1x PBS with 0.05% Tween 20 (Millipore)) and blocked with blocking buffer (1× PBS with 2% BSA (Sigma) and 0.05% Tween 20 (Millipore)) for 1 h at RT. The block buffer was removed and plates were then incubated with 0.625 µg/ml biotinylated HIV gp140 ConC_SOSIP (Janssen) for 1 h at RT. Rabbit sera, ConC standard serum, PGT145 and 447-52D mAbs (positive controls; Janssen and Polymun respectively), and CR9056 mAb (negative control; Janssen) were diluted in blocking buffer and 4-fold dilution series starting at 1:50 was performed for the sera and 5-fold dilution series starting at 40 µg/ml for the mAbs. The plates were then washed in wash buffer and the sera or mAbs were incubated with the coated Ag for 1 h at RT. Further washes, secondary antibody incubation and measurement of luminescence were performed as described above for the direct coat ELISA. A nonlinear regression curve fit with 4-logistic parameters based on the average of duplicates linked to the dilution on the original scale was used to generate response curves. The REP and LLOD were determined as described above for the direct coat ConC ELISA. All samples were run in duplicates.

## Determination of off-target antibody responses against the His-Tag with anti-His ELISA

A direct coat anti-His ELISA was used to assess the levels of off-target non-Env antibody responses. For this, 10 µg/ml FL-HA_H1 Brisbane His-Tag protein (Janssen) was coated onto 96 well plates (Perkin Elmer) in Carbonate/bi-carbonate buffer 0.05 M pH 9.6 (Janssen) at 4 °C overnight. The plates were washed with PBST wash buffer (1x PBS with 0.05% Tween 20 (Millipore)) and blocked with blocking buffer (1x PBS pH 7.4 with 2% Skim milk (BD Life Sciences)) for 1 h at RT. Rabbit and control sera were diluted in blocking buffer and 4-fold dilution series starting at 1:50 was performed. The block buffer was removed from the plates and the sera were incubated with the coated Ag for 1 h at RT. Next, plates were washed in wash buffer and incubated with 1:5,000 Goat anti-rabbit IgG HRP (Jackson ImmunoResearch) for 1 h at RT. The plates were washed in wash buffer and incubated with ECL solution by mixing Substrate A Luminol/enhance solution and Substrate B Peroxide solution (Bio-Rad) for 25 min at RT in the dark. Luminescence was measured using the BioTek Synergy Neo plate reader. Raw data from the luminescence reader is imported into an R script (R version 3.6.1) which calculates endpoint antibody binding titers: First a 4PL curve is fitted per sample. For this, 100,000 luminescence (LUM) units are added to all results in order to better fit a curve to the raw data and make the variance of measurements with low luminescence comparable to the variance observed of measurements with high luminescence. The R script then calculates where the fitted curve intersects the set threshold of 160,000 LUM (60,000 LUM without the 100,000 LUM addition), which corresponds to a low-level response usually not reached in naïve control sera. The R script calculates the reciprocal antibody titers from the concomitant serum dilution, which are provided here as log10 transformed endpoint titers (Endpoint titer Log10). The LLOD is defined as the reciprocal antibody titer of the serum start dilution used in the assay, for example, an LLOD endpoint titer of 2.00 Log10 when starting with a 1:100 serum dilution. All samples were run in duplicate.

## Determination of serum neutralizing antibody levels by TZM-bl assay

The neutralization capacity of rabbit sera against HIV-1 Env pseudo-typed viruses representing the global panel[52] and a selected clade C panel (Fig. 3), was performed using the luciferase-based TZM-bl assay[97,98]. Virus particles were generated by transfection of HEK-293T cells with pSG3Δenv along with pCDNA3.1 expressing the corresponding Env gene using FuGene 6 transfection reagent (Promega). The virus supernatants were harvested 48 h after transfection by centrifugation, filtered with a 0.45 µm filter and stored at -80 °C until use. TZM-bl cells were maintained in DMEM (Gibco) with 10% FCS (Gibco), 100 µg/ml Penicillin (Sigma)/Streptomycin (Thermo Fisher) at 37 °C and 5% $CO_2$. All virus productions were titrated using the TCID$_{50}$ method for infecting TZM-bl cells[98].

For measuring neutralizing activity in rabbit sera, samples were heat inactivated at 56 °C for 60 minutes and 3-fold serial dilutions of serum, with a 1:20 start dilution, were co-incubated with pseudovirus supernatants for 1 h at 37 °C, following which 80 μg/ml DEAE Dextran (Sigma), 2 nM Saquinavir (SQV)(NIH) and TZM-bl cells were added. After 3 days of incubation at 37 °C and 5% CO2, cells were lysed with pH 7.8 lysis buffer containing 3.3 %(v/w) Glycylglycine 1.8% (v/w) $MgSO_4$ anhydrous or 3.7 % (v/w) $MgSO_4$ heptahydrate 1.88 % (v/w) EGTA tetrasodium and 10% Triton X-100 in $H_2O$; and luciferase activity was determined by addition of Bright-Glo Luciferase mix (Promega) and measured on a luminescence microplate reader Glomax 96 (Promega). The background relative light units (RLUs) of non-infected cells were subtracted and then serum 50% inhibitory dilution ($ID_{50}$) was calculated as the log10 transformation of the dilution resulting in a 50% reduction in RLU compared with the untreated virus control as maximum signal for each sample. This was done by plotting a log (inhibitor) vs. response curve with variable slope using a non-linear 4 parameter curve fit of each dilution curve is created with a constraint bottom and top at 0% and 100% in GraphPad Prism. Values below the LLOD (serum $ID_{50}$ of 20) were assigned a value of 10 (1/2 the LLOD) and those above the upper limit of detection (ULOD) (serum $ID_{50}$ of 4860) were assigned a value of 4870 for plotting of graphs and statistical analysis. All samples were run in duplicate.

### HIV-1 Env sequence alignment and building of phylogenetic tree

HIV-1 Env sequence alignment using default program settings was generated using the CLC Main Workbench Software version 23.0.4. Computed distance and % identity of pairwise comparisons are shown in Supplementary fig. 4b. A Maximum Likelihood Phylogeny Tree was built in the CLC Main Workbench using the Neighbor Joining method without a bootstrap analysis.

### Neutralization fingerprinting for epitope mapping

Epitope mapping of serum neutralization was performed as previously described[53] with a few adjustments. Briefly, for antibodies, the reciprocals of the virus-neutralizing concentrations are used as titers. For sera and antibodies, the titers are scaled such that the limit of detection becomes one. Next, the titers are log-transformed. For each serum, a linear regression of the log-titer on the log-titers of all antibodies is done without intercept and with the constraint that all regression coefficients are non-negative. The regression coefficients per serum are interpreted as absolute mixture amounts of antibodies that best reflect the serum. For the sera in this study, the additional ranking step describe previously, resulting in relative mixture amounts is less desirable since the sera widely differ in breadth of neutralization. HIV-1 strains included for the Env epitope prediction of serum neutralization include: ZM233M.PB6, ZM215F. PB8, 25710-2.43, Ce1176_A3, Ce703010217_B6, 246-F3_C10_2, BJOX002000.03.2, CH119, 398F1, CNE8, TRO.11, X2278, X1632.S2-B10, CNE55. These strains were used based on a complete data set being available in the database[99] for the following reference broadly neutralizing antibodies: PGT145, PGDM1400, VRC26.25, PG16, PGT128, 10-1074, 2G12, PGT121, VRC01, N6, 3BNC117, CH103, VRC34.01, ACS202, 8ANC195, PGT151, 35O22, 10E8, 4E10 and 2F5. The prediction score result for all sera together is presented as a heatmap.

### Statistical analyses

For results from the neutralization, GMT was calculated as the geometric mean of serum $ID_{50}$ values against the 20 tier 2 strains per rabbit. % breadth was calculated as the fractions of strains neutralized out of the 20 tier 2 strains.

For the pairwise comparisons of vaccine groups in ELISA and neutralization readouts, an analysis-of-variance (ANOVA) was used, followed by Tukey's multiple comparisons test for measurements with a normal distribution, otherwise the Kruskal-Wallis test was used,

followed by Dunn's multiple comparisons test. The comparisons of ELISA titers between different time points per vaccine group were done with the paired t-test. The Spearman rank correlations was calculated between assays, together with an approximate p-value. Values below 0.05 p-value were considered as being significant.

### Reporting summary

Further information on research design is available in the Nature Portfolio Reporting Summary linked to this article.

## Data availability

All of the final data has been included in main figures or supplementary information. Any requests for protocols and reagents should be directed to the corresponding authors to be fulfilled under reasonable request. Source data are provided with this paper.

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

## Acknowledgements

We would like to thank J.E. Robinson for providing antibody 14e; Ludovic Renault at NeCEN for cryo-EM imaging. Vladimir van Hoek and Midia Khalifa for technical support; team members from Janssen Vaccines & Prevention for helpful discussions.

## Author contributions

Study design and planning A.K. K.V. G.H., L.R., N.S., D.M., R.Z. H.S. J.L., and F.W.; formal analysis A.K., K.V., and J.T; investigation A.K., J.T., K.F., J.V., A.P., S.B., J.A.B., and W-C.H.; resources L.R., N.S., J.F.L., and R.S; writing – original draft A.K., and K.V.; writing – reviewing and editing F.W., L.R., N.S., D.M., R.Z., and J.L. This work was funded by Janssen Vaccines & Prevention B.V.

## Competing interests

Parts of this work have been listed on the patent application by Janssen Vaccines & Prevention B.V and by POP Biotechnologies Inc. A.K., K.V., L.R., J.T., J.V., K.F, A.P., S.B., R.Z., H.S. and F.W. are employees of Janssen Vaccines & Prevention B.V. G.H., N.S., D.M., and J.L. are former employees of Janssen Vaccines & Prevention B.V. A.K., J.T., K.F., L.R., D.M., S.B., R.Z., H.S., J.L., and F.W. hold stock options in Johnson & Johnson. W-C.H. and J.L. are employees of and/or hold interest in POP Biotechnologies, Inc. All other authors declare no competing interests.
