## [Peer Review File · Nature Communications]

REVIEWER COMMENTS

Reviewer #1 (Remarks to the Author):

The manuscript by Koornneef et al titled “CoPoP liposomes decorated with stabilized clade C HIV-1 Env elicit tier 2 multi-clade neutralization in rabbits” has described the constitution and immunogenicity study of three clade C HIV-1 Env trimers, called ConC, sC4, C97ZA that are stabilized in a prefusion conformation and formulated in cobalt porphyrin-phospholipid (CoPoP) liposomes in rabbits. The authors formulated each Env trimer in three different CoPoP liposomes (CoP1, 2, and 3) with different lipid and adjuvant component, characterized the antigenicity of the Env-liposome complex by testing binding to a panel of reference antibodies and the stability in rabbit sera, and immunized rabbits in a sequential manner to evaluate immunogenicity. The immune sera from Env-CoP1 & 2 liposome immunized rabbits showed neutralization ID50 titers around 20, and breadth of 35% against a panel of 20 tier 2 viruses including that from the clade C panel and the global panel (cross-clades) after three immunizations within 18 weeks. The Env-CoP2 immune rabbits were then boosted with a mixture of Ad6 viruses expressing membrane bound version of Env with background of ConC, sC4, C97ZA on week 34. The resultant immune sera showed boosted Env binding titer as well as virus neutralization titers similar with the antibody responses stimulated by Env-CoP2 liposomes previously. 6-10 sequential immunizations with Env trimer formulated in liposomes or LNP formulated mRNA encoding HIV VLP prime and sequential boosters with Env trimers have been reported to resulted in sporadic cross-neutralizing antibody responses against tier 2 viruses in rabbits and non-human primates (Dubrovskaya et al., 2019, *Immunity* 51, 915–929; Zhang et al., 2021, *Nat Med* 27, 2234-2245) previously. The observation of moderate cross-clade tier 2 virus neutralization achieved by sequential Env-CoP liposome immunizations in this study corroborates the finding in previous studies and is promising. Overall the data are solid and the manuscript is well written. Some points are listed for the authors to clarify.

Major point

(1) To characterize the antigenicity of the Env-liposomes, a few bNAbs have been used without any CD4 binding site (CD4bs)-specific bNAbs. Do they have decent binding to CD4bs bNAbs (VRC01, N6)? What about the fusion peptide specific bNAbs (VRC34, ACS202)?

(2) Given the promising breadth of neutralization displayed by the immune sera, what are the Env neutralization epitopes targeted by these immune sera? Are they specific for conserved epitopes such as the CD4 binding site, fusion peptide, or the trimer apex? Is there a predominantly targeted epitope or the sera contain multiple specificities? The authors need to analyze the serum neutralization specificity using methods described in Dubrovskaya et al., 2019, *Immunity* 51, 915–929; Georgiev et al., 2013, *Science*, 340, 751-6.

(3) The use of Ad26 vectored Envs in studying recalling immune response is not clearly justified. Would Envs displayed on liposomes do the same job? There is no side by side comparison in this study unfortunately. The author should state that further studies are needed to test if there is beneficial effect for using Env in Ad26 vectors. Please consider revising the sentence stating “A

heterologous regimen consisting of protein and genetically encoded immunogen could thus be a promising tool for an HIV-1 vaccine” in lines 323-324.

(4) The authors should consider listing the statistically significant difference ($p < 0.05$) in Fig. 3b, 4d.

Minor points

(1) Supplementary figure 5b, what is the pacific blue stain for (cell death)? Is VRC26 labeled with APC? The method stated goat anti-human IgG Alexa Fluor 647 was used as secondary antibody. Please clarify.

Reviewer #2 (Remarks to the Author):

Koornneef and colleagues report on a promising new strategy for the design of HIV-1 Env-based immunogens using native-like trimers stably and directionally bound to CoPoP liposomes. The virus-like size of these particles and the abundance of Env displayed on their surface make the new immunogens attractive candidate vaccines. To verify immunogenicity, the authors sequentially immunized rabbits with three different trimers (ConC-CoPoP, sC4-CoPoP and C97ZA-CoPoP) inoculated with Adjuvax as an adjuvant at three time points (week 0, week 8 and week 16). They observed robust anti-Env antibody induction and some degree of broad-spectrum neutralization, albeit at low level and not consistent in all animals. These results are nevertheless interesting and should be replicated in a more relevant animal model such as macaques.

Major points to be addressed:

1. The characterization of the immunogens is incomplete. In Fig. 1, only a limited panel of bNabs was tested, with only two of them being trimer specific. The authors should test at least one or more bNabs against other critical neutralization sites such as the CD4-binding site (eg, VRC01, N6, 3BNC117, etc.) and the gp120/gp41 interface region (eg, PGT151, 35O22, etc.). It is critical to know if these regions are correctly folded and displayed on these immunogens.
2. Another recommended characterization is to perform NSEM or cryoEM analysis of the liposome-trimer particles. This will corroborate the claim that the particles are homogeneous in size and decorated with a rich complement of Env spikes.
3. Despite the low titers and the inconsistencies among various animals, the neutralization breadth elicited by immunization seems to be real (Fig. 3). The authors should be careful in defining all the Envs included in the global panel as tier 2, since at least 3 of them are relatively easy to neutralize, likely due to enhanced exposure of the V3 loop, and their real tier categorization is debated.

4. The authors should attempt to identify the epitope targets of such broad neutralization. In the absence of a detailed clonal analysis of the elicited B-cell repertoire, two feasible approaches include bNAb-competition experiments and neutralization of Env mutants lacking specific epitopes.

5. The rabbit sera should be tested for antibodies to the trimer base. Although the trimer base in these immunogens should be largely hidden by directional conjugation with the liposomes, it is still possible that such antibodies be elicited. This is an important question to address.

6. The major drawback of the immunization schema employed in this study is that there was apparently no rational selection of the sequential Envs used. Especially if they plan to move to a more relevant animal model such as macaques, the authors should consider employing a rational sequence of immunogens to shepherd the affinity maturation of highly potent bNAbs. They should select a priming immunogen with the ability to engage efficiently bNAb precursors and then follow-up with progressively more closed trimers, possibly from different clades, to focus the antibody maturation on shared epitopes, which are by definition bNAb epitopes.

Reviewer #3 (Remarks to the Author):

In this manuscript, Koornneef et al. described an HIV-1 vaccine candidate that consists of cobalt porphyrin-phospholipid (CoPOP) liposomes decorated with stabilized clade C HIV-1 envelope glycoprotein (Env) trimers. This CoPOP liposome-based HIV-1 vaccine elicited cross-clade neutralizing antibody (NAb) responses in rabbits that can neutralize up to 35% breadth in a panel of twenty tier 2 strains. The authors also showed that the peak NAb titers can be recalled by a heterologous immunization with an Ad26-vectored HIV-1 Env vaccine. Despite the positive data, the logic, rationale, and significance of this study must be clarified and improved. I suggest the authors conduct more experiments to demonstrate the unique properties and advantages of this CoPOP liposome vaccine in comparison with previously reported vaccines that utilize liposomes and cobalt conjugation as well as other delivery systems to present stabilized HIV-1 Env trimers. Below are my comments that may help authors improve the quality of this manuscript. Overall, this work may be more appropriate for publication in a specialty journal.

Major comment #1:

The logic, rationale, and significance of this study must be clarified. Richard Wyatt's group has published multiple papers on the use of liposomes to present SOSIP or NFL trimers for HIV-1 vaccine development (Ingale et al., 2016, Cell rep., PMID: 27210756; Bale et al., 2017, J. Vir., PMID: 28592540; Martinez-Murillo et al., 2017, Immunity, PMID: 28514687; Dubrovskaya et al., 2019, Immunity, PMID: 31732167). For example, SOSIP and NFL trimers were conjugated onto liposomes through synthetic nickel (Ingale et al., 2016, Cell rep., PMID: 27210756). NFL trimers of multiple clades, e.g., BG505 (clade A), 16055 (clade C), and SC422 (clade B) trimers, were covalently

conjugated onto cobalt- and maleimide-containing liposomes (Bale et al., 2017, *J. Vir.*, PMID: 28592540). These trimer-presenting liposomes were shown to stimulate B cells, induce robust germinal centers, and elicit cross-clade NABs in rabbits (Dubrovskaya et al., 2019, *Immunity*, PMID: 31732167). This approach has been successfully elicited CD4bs NABs with considerable breadth and an NAb targeting the gp120-gp41 interface with 87% breadth. Since HIV-1 Env-liposomes have been extensively studied in rabbits with so many publications, the authors need to explain why their work should be considered different and significant. It is also worth pointing out that all studies of cross-NAB elicitation using Env-liposomes were reported in rabbits. The rabbit immune system uses a unique gene swapping mechanism instead of somatic hypermutation to generate functional antibodies. For this reason, it is easier to elicit HIV-1 NABs in rabbits than mice and primates. However, the rabbit immunogenicity has very little bearing on what the vaccine may do in other animal models using a similar immune system to humans. The authors are encouraged to at least test another animal model, e.g., wild type mice.

Major comment #2:

The authors focused on clade C HIV-1 Env trimers in this study, but clade C Env-presenting liposomes have been extensively studied (Bale et al., 2017, *J. Vir.*, PMID: 28592540; Martinez-Murillo et al., 2017, *Immunity*, PMID: 28514687). What is the rationale of choosing clade C and not the other clades for immunogen development?

Major comment #3:

Additional characterizations of Env-CoPoP liposomes are needed. For example, the authors used bNAbs to target the epitopes of their Env designs and claimed that these Envs are in a prefusion-closed trimer structure. Negative stain electron microscopy (nsEM) is the minimum requirement for validation of such a claim. In addition, the authors reported that their CoPoP liposomes present a dense array of stably attached HIV-1 Env trimers and these trimers are monodispersed. It was rather disappointing to see that they only used DLS to characterize the physicochemical properties of liposome presented Env trimers. Again, in this case, nsEM images would be the minimum requirement to show the morphology of Env-CoPOP nanoparticles.

Major comment #4:

The authors stated that "As a control, Env-Ni-NTA liposomes that have been shown to dissociate in serum were prepared in a similar fashion to CoPoP liposomes." In one of the Wyatt's papers (Bale et al., 2017, *Journal of Virology*, PMID: 28592540), they reported that both cobalt and cysteine coupling resulted in a high-density array of NFL trimers that was stable in 20% mouse serum, whereas the nickel-conjugated trimers were not stable under this condition. In addition, Bale et al. reported that this covalent coupling prevents release of the trimers prior to recognition by B cells and masks a non-neutralizing determinant located at the trimer bottom. The authors should not use Env-Ni-NTA

liposomes as a control in the current study. Instead, they should use cobalt-linked liposomes as a fair comparison to their CoPoP liposomes.

Major comment #5:

In Figure 3, the geometric mean titers (GMT) of serum neutralization were calculated against 20 Env pseudotyped viruses using a TZM-bl neutralization assay. The ID50 titers against the 20 viruses varied significantly, in the range of 0 to over 1000. How can neutralization GMT across all 20 viruses be used to determine the potency against completely different strains?

Major comment #6:

Why did the authors only use the Ad26.RnS.Env cocktail as a boost in heterologous vaccination regimens? This choice did not make sense, as vector vaccines are known to be less advantageous than protein vaccines in antibody responses. The authors should include an Env-CoPoP liposome vaccine as a booster for comparison. It is likely that the Env-presenting liposome will generate a stronger recall response, further expanding the breadth of the NAb responses.

Major comment #7:

It is unclear why the authors chose the Env-CoPOP2 group and not the Env-CoPOP1 group for heterologous vaccination. The Env-CoPOP1 group appeared to show greater potency and breadth against eight tier 2 clade C strains and other strains on the global panel, as shown in Figure 3a.

Major comment #8:

The authors reported that “the average breadth coverage of neutralizing responses elicited by the HIV-1 Env-decorated CoPoP particles was up to 35% against a multi-clade panel”. It would make sense to study which epitopes these NABs target, e.g., using some epitope-knockout mutant viruses.

Reviewer #1 (Remarks to the Author):

The manuscript by Koornneef et al titled “CoPoP liposomes decorated with stabilized clade C HIV-1 Env elicit tier 2 multi-clade neutralization in rabbits” has described the constitution and immunogenicity study of three clade C HIV-1 Env trimers, called ConC, sC4, C97ZA that are stabilized in a prefusion conformation and formulated in cobalt porphyrin-phospholipid (CoPoP) liposomes in rabbits. The authors formulated each Env trimer in three different CoPoP liposomes (CoP1, 2, and 3) with different lipid and adjuvant component, characterized the antigenicity of the Env-liposome complex by testing binding to a panel of reference antibodies and the stability in rabbit sera, and immunized rabbits in a sequential manner to evaluate immunogenicity. The immune sera from Env-CoP1 & 2 liposome immunized rabbits showed neutralization ID50 titers around 20, and breadth of 35% against a panel of 20 tier 2 viruses including that from the clade C panel and the global panel (cross-clades) after three immunizations within 18 weeks. The Env-CoP2 immune rabbits were then boosted with a mixture of Ad6 viruses expressing membrane bound version of Env with background of ConC, sC4, C97ZA on week 34. The resultant immune sera showed boosted Env binding titer as well as virus neutralization titers similar with the antibody responses stimulated by Env-CoP2 liposomes previously. 6-10 sequential immunizations with Env trimer formulated in liposomes or LNP formulated mRNA encoding HIV VLP prime and sequential boosters with Env trimers have been reported to resulted in sporadic cross-neutralizing antibody responses against tier 2 viruses in rabbits and non-human primates (Dubrovskaya et al., 2019, Immunity 51, 915–929; Zhang et al., 2021, Nat Med 27, 2234-2245) previously. The observation of moderate cross-clade tier 2 virus neutralization achieved by sequential Env-CoP liposome immunizations in this study corroborates the finding in previous studies and is promising. Overall the data are solid and the manuscript is well written. Some points are listed for the authors to clarify.

We thank the reviewer for their time in helping us improve our study and evaluating our work as “*data are solid and the manuscript is well written*”. We have addressed specific points raised by the reviewer below.

Major point

1. To characterize the antigenicity of the Env-liposomes, a few bNAbs have been used without any CD4 binding site (CD4bs)-specific bNAb. Do they have decent binding to CD4bs bNAbs (VRC01, N6)? What about the fusion peptide specific bNAbs (VRC34, ACS202)?

We thank the reviewer for this comment and have now extended the bNAb panel that is used to characterize the Envs with CD4bs bNAbs (VRC01 and 3BNC60), and with bNAbs binding at the gp120-gp41 interface and fusion peptide (35O22, PGT151, VRC34). We show *in vitro* binding to the V1/V2 apex, V3 region, the gp120-gp41 interface and the fusion peptide in the absence of non-bNAb binding. However, sC4 and C97ZA showed reduced binding to the CD4bs-targeting Abs. These Envs contain a disulfide that prevents CD4 induction and as a consequence these variants are recognized less by bNAbs against the CD4bs,

which was reported in *Nguyen et al, Journal of Virology 2019 doi:10.1128/JVI.00304-19*. The data is now included in figure 1a and discussed in the manuscript in lines 101-104 and 296-300. Additionally, the immune serum neutralization-based epitope prediction, which is now presented in Fig. 3c, revealed antibody responses against the variable loops, CD4bs, fusion peptide, interface and the MPER, corroborating the broad antigenicity of the vaccine antigens.

2. Given the promising breadth of neutralization displayed by the immune sera, what are the Env neutralization epitopes targeted by these immune sera? Are they specific for conserved epitopes such as the CD4 binding site, fusion peptide, or the trimer apex? Is there a predominantly targeted epitope or the sera contain multiple specificities? The authors need to analyze the serum neutralization specificity using methods described in Dubrovskaya et al., 2019, *Immunity* 51, 915–929; Georgiev et al., 2013, *Science*, 340, 751-6.

Thank you for this comment. We have now performed epitope mapping of the sera using the technique of neutralization fingerprinting based on *Georgiev et al Science 2013 doi: 10.1126/science.1233989*.

Based on the neutralization fingerprinting data, we see predicted reactivity of the sera from all three groups primarily against the fusion peptide and the gp41-gp120 interface. Of note, groups 1 and 2, which showed superior neutralization breadth, display greater reactivity that maps to the variable loops, CD4bs and MPER versus group 3, which could perhaps explain the greater breadth seen in these groups against the global and clade C panels. We have included these data in the manuscript in figure 3c and discussed these observations in lines 213-219 and 326-330.

3. The use of Ad26 vectored Envs in studying recalling immune response is not clearly justified. Would Envs displayed on liposomes do the same job? There is no side by side comparison in this study unfortunately. The author should state that further studies are needed to test if there is beneficial effect for using Env in Ad26 vectors. Please consider revising the sentence stating “A heterologous regimen consisting of protein and genetically encoded immunogen could thus be a promising tool for an HIV-1 vaccine” in lines 323-324.

Thank you for raising this point and we agree that a head-to-head comparison between Ad26 boost and CoPoP would be beneficial. However, in the present study, splitting up of a group into two separate boost immunization arms was not feasible at sufficient statistical power based on the resulting group size and based on the variation in serum neutralizing activity observed within a group. Due to the real-world success of heterologous vaccinations during the COVID pandemic, our focus for this study was to test a heterologous delivery system in the form of membrane-bound stabilized Envs encoded by the Ad26 vector. Moreover, membrane-bound Env has been demonstrated to display a more authentic glycosylation in Cao

et al Nature communications 2018 doi:10.1038/s41467-018-06121-4, therefore we hypothesized that the Ad26 booster might further drive affinity maturation towards bnAbs. Follow up work would indeed require the assessment of whether a CoPoP boost improves the potency and breadth of the response induced by the three-dose primary CoPoP regimen. Lines 345-348 have now been modified in the text to account for this point.

4. The authors should consider listing the statistically significant difference ($p < 0.05$) in Fig. 3b, 4d.

Thank you for this suggestion, the p values are included in supplementary table 1, to reduce crowding in figures we refer to the table for p values.

Minor points

1. Supplementary figure 5b, what is the pacific blue stain for (cell death)? Is VRC26 labeled with APC? The method stated goat anti-human IgG Alexa Fluor 647 was used as secondary antibody. Please clarify.

Thank you for raising this point. We apologize for the confusion, which likely stems from the label that is automatically given by the FACSDiva software. We have corrected the supplementary figure labels. The 'Pacific blue' label is now adjusted to 'Violet live/dead'. VRC26 was measured via a secondary goat anti-human IgG alexa fluor 647 as stated in the methods section. We have now adjusted the 'APC' label to 'VRC26'.

Reviewer #2 (Remarks to the Author):

Koornneef and colleagues report on a promising new strategy for the design of HIV-1 Env-based immunogens using native-like trimers stably and directionally bound to CoPoP liposomes. The virus-like size of these particles and the abundance of Env displayed on their surface make the new immunogens attractive candidate vaccines. To verify immunogenicity, the authors sequentially immunized rabbits with three different trimers (ConC-CoPoP, sC4-CoPoP and C97ZA-CoPoP) inoculated with Adjuvax as an adjuvant at three time points (week 0, week 8 and week 16). They observed robust anti-Env antibody induction and some degree of broad-spectrum neutralization, albeit at low level and not consistent in all animals. These results are nevertheless interesting and should be replicated in a more relevant animal model such as macaques.

We thank the reviewer for their careful assessment of our manuscript and finding our results “*interesting*”. We have addressed the specific comments below.

Major points to be addressed:

1. The characterization of the immunogens is incomplete. In Fig. 1, only a limited panel of bNabs was tested, with only two of them being trimer specific. The authors should test at least one or more bNabs against other critical neutralization sites such as the CD4-binding site (eg, VRC01, N6, 3BNC117, etc.) and the gp120/gp41 interface region (eg, PGT151, 35O22, etc.). It is critical to know if these regions are correctly folded and displayed on these immunogens.

We thank the reviewer for the comment and have also addressed this point in our response to reviewer 1 as follows: we have now extended the bNAb panel that is used to characterize the Envs with CD4bs bNabs (VRC01 and 3BNC60) and with bNabs binding at the gp120-gp41 interface and fusion peptide (35O22, PGT151, VRC34). We demonstrate *in vitro* binding to the V1/V2 apex, V3 region, the gp120-gp41 interface and the fusion peptide, in the absence of non-bNAb binding. However, we see a lower response to the CD4bs, particularly for the sC4 and C97ZA Envs. These Envs contain a disulfide that prevents CD4 induction and as a consequence these variants are recognized less by bNabs against the CD4bs, which was reported in *Nguyen et al, Journal of Virology 2019 doi:10.1128/JVI.00304-19*. The data are now included in figure 1a and discussed in the manuscript in lines 101-104 and 296-300. Additionally, the immune serum neutralization-based epitope prediction, which is now presented in Fig. 3c, revealed antibody responses against the variable loops, CD4bs, fusion peptide, interface and the MPER, corroborating the broad antigenicity of the vaccine antigens.

2. Another recommended characterization is to perform NSEM or cryoEM analysis of the liposome-trimer particles. This will corroborate the claim that the particles are homogeneous in size and decorated with a rich complement of Env spikes.

Thank you for this suggestion. Cryo-EM analysis has been performed on the Env-CoPoP liposomes and images are now reported in Figure 1d and discussed in lines 123-125. The images show spherical particles of approximately 100 nm diameter in size, displaying a dense decoration of Env spikes on the liposome surface.

3. Despite the low titers and the inconsistencies among various animals, the neutralization breadth elicited by immunization seems to be real (Fig. 3). The authors should be careful in defining all the Envs included in the global panel as tier 2, since at least 3 of them are relatively easy to neutralize, likely due to enhanced exposure of the V3 loop, and their real tier categorization is debated.

We thank the reviewer for the comment. We do agree with the reviewer that there are sometimes discussions on the Tier classification of standard HIV-1 panels. Currently, the Los Alamos database lists 10 out of 12 global panel strains as Tier 2, with two other strains being ambiguous as either i.) Tier 1B or Tier 2 and ii.) Tier 2 or Tier 3 (highest level of difficulty to neutralize). Therefore, in order to stay with standardized terminology used in published scientific studies and maintain consistency, the tiering for the global panel strains has been defined as Tier 2 as per the original article from *deCamp et al., Journal of Virology 2014 doi: 10.1128/JVI.02853-13*.

4. The authors should attempt to identify the epitope targets of such broad neutralization. In the absence of a detailed clonal analysis of the elicited B-cell repertoire, two feasible approaches include bNAb-competition experiments and neutralization of Env mutants lacking specific epitopes.

Thank you very much for this comment. Please also refer to our responses to reviewers 1 and 3 on this topic which are as follows: we have now performed epitope mapping of the sera using the technique of neutralization fingerprinting based on *Georgiev et al Science 2013 doi: 10.1126/science.1233989*. Based on the neutralization fingerprinting data, we see predicted reactivity of the sera from all three groups primarily against the fusion peptide and the gp41-gp120 interface. Of note, groups 1 and 2, which showed superior neutralization breadth compared with group 3, display greater reactivity that maps to the variable loops, CD4bs and MPER, which could perhaps explain the greater breadth seen in these groups against the global and clade C panels. We have included the data in the manuscript in figure 3c and discussed these observations in lines 213-219 and 326-330.

5. The rabbit sera should be tested for antibodies to the trimer base. Although the trimer base in these immunogens should be largely hidden by directional conjugation with the liposomes, it is still possible that such antibodies be elicited. This is an important question to address.

We thank the reviewer for this comment and agree that the trimer base of HIV-1 Env may become exposed to the immune system as a result from detachment of trimers from the liposome surface or degradation of the liposomes *in vivo*. We have used the C-terminal His-tag present on the Env immunogens as a proxy to interrogate antibody responses to the base of the trimer, which is reported in Figure 2b. We can detect a His-tag response above LLOD, albeit at 9-fold lower levels than observed with a Ni-NTA particle-based immunization schedule using identical His-tagged antigens (Supplementary figure 3b). Therefore, response to the trimer base, though not completely absent, is minimal and does not appear to hamper the Env immunogen specific response.

6. The major drawback of the immunization schema employed in this study is that there was apparently no

rational selection of the sequential Envs used. Especially if they plan to move to a more relevant animal model such as macaques, the authors should consider employing a rational sequence of immunogens to shepherd the affinity maturation of highly potent bNAbs. They should select a priming immunogen with the ability to engage efficiently bNAb precursors and then follow-up with progressively more closed trimers, possibly from different clades, to focus the antibody maturation on shared epitopes, which are by definition bNAb epitopes.

We thank the reviewer for this comment and apologize that this point was not made clear in the text. The HIV-1 Env strains used in the study were administered sequentially in the specific order consisting of ConC → sC4 → C97ZA based on the strategy of guiding the immune system to generate bNAbs by increasing the mutational distance of HIV-1 Env in relation to transmitter founder virus sequences at each immunization step. The rationale is based on several investigations on the sequence evolution of Env in HIV-1 infected individuals that have developed bNAb responses i.e. elite neutralizers, reported in *Liao et al. Nature 2013 doi: 10.1038/nature12053*, *Andrabi et al., Immunity 2015 doi: 10.1016/j.immuni.2015.10.014*, *Bhiman et al., Nat Med 2015 doi: 10.1038/nm.3963*, *Scharf et al. eLife 2016 doi: 10.7554/eLife.13783*. Based on these studies, the evolution of key residues crucial for bNAb generation (e.g., amino acid positions 156, 158, 160, 162, 166-173, 175, 275-281, 316, 361-369) was examined to determine suitability of the Env immunogens with the aim to mimic the sequence evolution needed to generate bNAbs. The consensus clade C sequence (ConC) emerged as the best candidate for prime immunization and this outcome also agrees with the finding that transmitted founder virus sequences are closer to consensus sequences in *Carlson et al., Science 2015: doi: 10.1126/science.1254031*. The second immunogen selected by us, sC4, resembles a sequence found later in infection, and C97ZA corresponds to more mutated sequences and was hence used last. We have also elaborated on this rationale in the text to describe this more clearly in lines 162-165.

Reviewer #3 (Remarks to the Author):

In this manuscript, Koornneef et al. described an HIV-1 vaccine candidate that consists of cobalt porphyrin-phospholipid (CoPOP) liposomes decorated with stabilized clade C HIV-1 envelope glycoprotein (Env) trimers. This CoPOP liposome-based HIV-1 vaccine elicited cross-clade neutralizing antibody (NAb) responses in rabbits that can neutralize up to 35% breadth in a panel of twenty tier 2 strains. The authors also showed that the peak NAb titers can be recalled by a heterologous immunization with an Ad26-vectored HIV-1 Env vaccine. Despite the positive data, the logic, rationale, and significance of this study must be clarified and improved. I suggest the authors conduct more experiments to demonstrate the unique properties and advantages of this CoPOP liposome vaccine in comparison with previously reported vaccines that utilize liposomes and cobalt conjugation as well as other delivery systems to present stabilized

HIV-1 Env trimers. Below are my comments that may help authors improve the quality of this manuscript. Overall, this work may be more appropriate for publication in a specialty journal.

We thank the reviewer for the feedback and helping us improve our study. In the revised manuscript, we have better highlighted the significance of the study, in addition to providing new data that improves our understanding of the CoPoP HIV-1 vaccine. The individual concerns from the reviewer are addressed below.

Major comment #1:

The logic, rationale, and significance of this study must be clarified. Richard Wyatt's group has published multiple papers on the use of liposomes to present SOSIP or NFL trimers for HIV-1 vaccine development (Ingale et al., 2016, Cell rep., PMID: 27210756; Bale et al., 2017, J. Vir., PMID: 28592540; Martinez-Murillo et al., 2017, Immunity, PMID: 28514687; Dubrovskaya et al., 2019, Immunity, PMID: 31732167). For example, SOSIP and NFL trimers were conjugated onto liposomes through synthetic nickel (Ingale et al., 2016, Cell rep., PMID: 27210756). NFL trimers of multiple clades, e.g., BG505 (clade A), 16055 (clade C), and SC422 (clade B) trimers, were covalently conjugated onto cobalt- and maleimide-containing liposomes (Bale et al., 2017, J. Vir., PMID: 28592540). These trimer-presenting liposomes were shown to stimulate B cells, induce robust germinal centers, and elicit cross-clade Nabs in rabbits (Dubrovskaya et al., 2019, Immunity, PMID: 31732167). This approach has been successfully elicited CD4bs Nabs with considerable breadth and an NAb targeting the gp120-gp41 interface with 87% breadth. Since HIV-1 Env-liposomes have been extensively studied in rabbits with so many publications, the authors need to explain why their work should be considered different and significant. It is also worth pointing out that all studies of cross-NAb elicitation using Env-liposomes were reported in rabbits. The rabbit immune system uses a unique gene swapping mechanism instead of somatic hypermutation to generate functional antibodies. For this reason, it is easier to elicit HIV-1 Nabs in rabbits than mice and primates. However, the rabbit immunogenicity has very little bearing on what the vaccine may do in other animal models using a similar immune system to humans. The authors are encouraged to at least test another animal model, e.g., wild type mice.

We thank the reviewer for these comments. We kindly refer the reviewer to our response below to major comment 4, where we discuss the rationale for the use of CoPoP, and we apologize that this point was not made clear in the text. Points raised towards the choice of animal model and neutralization response are addressed here:

Rabbits have been used extensively in the field of HIV-1 vaccines to study the induction of bNAb responses due to their ability to form antibodies with long CDRH3s required for bNAb function like in humans. Few examples of such studies include *Klasse et al., Plos Pathogens 2016 doi:10.1371/journal.ppat.1005864,*

Sanders et al., Science 2020 DOI: 10.1126/science.aac4223, Mohan et al, 2018 DOI:10.1038/s41598-018-25960-1, Bianchi et al, Immunity 2018 <https://doi.org/10.1016/j.immuni.2018.07.009>. While we do agree that this small animal model has the disadvantage of its genetic distance from humans, it is an ideal small animal model when the study focuses on bNAb responses, and when assessing novel immunogens and formulations for their capacity to induce bNAb responses. Mice in contrast are both genetically distant and unable to induce antibodies with long CDRH3s for studying bNAb induction. A next step towards clinical development of the CoPoP HIV-1 vaccine would be testing in NHPs since they provide the closest resemblance to humans. The challenges of using large animals limits their use as first choice for preclinical evaluation and thus we think our choice of rabbits was justified within the scope of the current study.

With regard to the comment on the neutralization breadth seen in the rabbits, please note that the breadth of 35% is the % of viruses neutralized by the rabbit serum and not an individual monoclonal antibody. In the reference mentioned by the reviewer i.e. *Dubrovskaya et al., 2019, Immunity doi:10.1016/j.immuni.2019.10.008*, a single mAb shows 87% breadth, however, the serum responses have lower coverage due to the polyclonal nature of whole serum vs. a single potent monoclonal antibody. Therefore, we believe that the HIV-1 Env-CoPoP vaccine platform is a positive step forward in the HIV-1 vaccine field to induce multi-clade broad serum neutralizing antibody response.

Major comment #2:

The authors focused on clade C HIV-1 Env trimers in this study, but clade C Env-presenting liposomes have been extensively studied (Bale et al., 2017, *J. Vir.*, PMID: 28592540; Martinez-Murillo et al., 2017, *Immunity*, PMID: 28514687). What is the rationale of choosing clade C and not the other clades for immunogen development?

Thank you for your question. We have focused our efforts on clade C Envs based on the widespread occurrence of subtype C, accounting for almost half of all HIV-1 infections globally. Subtype C is followed in prevalence by subtype B, which is responsible for 12% of infections (Hemelaar et al., *The Lancet* 2018 [doi.org/10.1016/S1473-3099\(18\)30647-9](https://doi.org/10.1016/S1473-3099(18)30647-9)). In earlier work on the repair and stabilize (RnS) method, it was observed that clade C Envs show more optimal folding of closed Envs compared to clade B Envs and hence, the success rate of the RnS method was highest for clade C Envs (87%) and reached only 55% for the clade B strains (Rawi et al. *Cell Rep* 2020 [doi: 10.1016/j.celrep.2020.108432](https://doi.org/10.1016/j.celrep.2020.108432)). Since a well-folded closed trimer is more likely to induce broad neutralizing antibodies, we therefore chose clade C Env for current immunogen development.

Major comment #3:

Additional characterizations of Env-CoPoP liposomes are needed. For example, the authors used bNAbs to target the epitopes of their Env designs and claimed that these Envs are in a prefusion-closed trimer structure. Negative stain electron microscopy (nsEM) is the minimum requirement for validation of such a claim. In addition, the authors reported that their CoPoP liposomes present a dense array of stably attached HIV-1 Env trimers and these trimers are monodispersed. It was rather disappointing to see that they only used DLS to characterize the physicochemical properties of liposome presented Env trimers. Again, in this case, nsEM images would be the minimum requirement to show the morphology of Env-CoPOP nanoparticles.

We thank the reviewer for raising these points and for the suggestions. We have provided additional characterization of the Env-CoPoP liposomes by i) additional antigenicity characterization (figure 1a), and ii) cryo-EM imaging of the Env-CoPoP liposomes, showing that the liposomes are displaying trimeric Env spikes (figure 1d).

Regarding the comment on the prefusion-closed trimer structure of Env, we would kindly refer the reviewer to the crystal structure of a ConC protein that was published by Rutten et al. in *Cell Reports*, 2018 doi: 10.1016/j.celrep.2018.03.061, where the prefusion-closed trimer conformation of the stabilized protein was demonstrated. The Env designs used for the Env-CoPoP liposomes are based on the introduction of stabilizing mutations from Rutten et al *Cell Reports*, 2018.

Major comment #4:

The authors stated that “As a control, Env-Ni-NTA liposomes that have been shown to dissociate in serum were prepared in a similar fashion to CoPoP liposomes.” In one of the Wyatt’s papers (Bale et al., 2017, *Journal of Virology*, PMID: 28592540), they reported that both cobalt and cysteine coupling resulted in a high-density array of NFL trimers that was stable in 20% mouse serum, whereas the nickel-conjugated trimers were not stable under this condition. In addition, Bale et al. reported that this covalent coupling prevents release of the trimers prior to recognition by B cells and masks a non-neutralizing determinant located at the trimer bottom. The authors should not use Env-Ni-NTA liposomes as a control in the current study. Instead, they should use cobalt-linked liposomes as a fair comparison to their CoPoP liposomes.

We thank the reviewer for this suggestion and would like to elaborate on our choice for CoPoP as a testing platform and Ni-NTA as a comparator.

In the referenced study of Bale et al. *J Virol* 2017: doi: 10.1128/JVI.00443-17, cobalt did not give satisfactory results *in vivo* and indeed a conclusion of that work was covalent coupling was superior to non-covalent coupling. Co-NTA-lipid has been previously tested and was found unable to stably sequester his-tagged proteins in *Shao et al., Nature Chemistry* 2015 doi:10.1038/nchem.2236. This was further studied in depth in *Federizon et al., Pharmaceutics* 2021 DOI: 10.3390/pharmaceutics13010098, where it was

demonstrated that, while several cobalt-chelate complexes could bind his-tagged proteins in solution, only CoPoP gave rise to stable protein binding and favorable immune responses. Furthermore, molecular dynamics simulations showed that by sequestering cobalt in the bilayer with CoPoP, it is relatively shielded from water interaction in the bilayer compared to Co-NTA-lipid, thus providing a mechanism of the improved binding stability. In addition, there is rationale for making use of a cobalt chelating technology that has been shown to be safe and immunogenic in human vaccines, which is the case for CoPoP, but not Co-NTA (Lovell et al., BMC Med 2022: doi:10.1186/s12916-022-02661-1).

The suggestion to use Co-NTA as a comparator is an interesting one, but we believe there are reasons why Ni-NTA is a more suitable comparator. The main reason is that there are numerous prior works that make use of Ni-NTA liposomes for sequestration of his-tagged antigens. Examples include *Pejawar-Gaddy et al., Bioconjugate Chemistry 2014 doi: 10.1021/bc5002246*, *Ingale et al., Cell Reports 2016 doi: 10.1016/j.celrep.2016.04.078*, *Steichen et al., Immunity 2016 doi: 10.1016/j.immuni.2016.08.016*, *Krupka et al., PLoS One 2016 doi: 10.1371/journal.pone.0148497*, *Martinez-Murillo et al., Immunity 2017 doi: 10.1016/j.immuni.2017.04.021*, *Tokatlian et al., Scientific Reports 2018 doi: 10.1016/j.celrep.2016.04.078*, *Dubrovskaya et al., Immunity 2019 doi: 10.1016/j.immuni.2019.10.008*, while there is relatively little work on Co-NTA lipid vaccines. We add as a final point that the main scope of this study is on the vaccine efficacy of CoPoP liposomes. We have added wording in the introduction to describe better the unique properties and advantages of CoPoP liposomes, referring to earlier work of *Federizon et al., Pharmaceutics 2021 DOI: 10.3390/pharmaceutics13010098* and *Shao et al., Nature Chemistry 2015 doi:10.1038/nchem.2236* in lines 76-80 of the manuscript.

Major comment #5:

In Figure 3, the geometric mean titers (GMT) of serum neutralization were calculated against 20 Env pseudotyped viruses using a TZM-bl neutralization assay. The ID50 titers against the 20 viruses varied significantly, in the range of 0 to over 1000. How can neutralization GMT across all 20 viruses be used to determine the potency against completely different strains?

Thank you for this comment. It is common practice in the field to assess combined neutralization capacity of study subjects by using a geometric mean across different strains. A few selected references of such analyses include the Los alamos HIV database antibody tool, *Mishra et al., Nature Communications 2020 <https://doi.org/10.1038/s41467-020-18225-x>*, *Xu et al., Science 2017 10.1126/science.aan8630*, *Schommers et al., Cell 2020 <https://doi.org/10.1016/j.cell.2020.01.010>*, *Sok et al., Immunity 2016 <https://doi.org/10.1016/j.immuni.2016.10.033>*, *Schooten et al., Nature Communications 2022 <https://doi.org/10.1038/s41467-022-32208-0>*, *Georgiev et al., Science 2013 doi: 10.1126/science.1233989*. The use of such analyses gives us the possibility to compare potential candidates for further development.

Lines 222-229 in the text provide a brief explanation of this analysis for comparison of the CoPoP regimen within the current study.

Major comment #6:

Why did the authors only use the Ad26.RnS.Env cocktail as a boost in heterologous vaccination regimens? This choice did not make sense, as vector vaccines are known to be less advantageous than protein vaccines in antibody responses. The authors should include an Env-CoPoP liposome vaccine as a booster for comparison. It is likely that the Env-presenting liposome will generate a stronger recall response, further expanding the breadth of the NAb responses.

Thank you for raising this point. We have also addressed this point in our response to reviewer 1 as follows: we do agree that a head-to-head comparison between Ad26 boost and CoPoP would be beneficial. However, in the present study, splitting up of a group into two separate boost immunization arms was not feasible at sufficient statistical power based on the resulting group size and based on the variation in serum neutralizing activity observed within a group. Due to the real-world success of heterologous vaccinations during the COVID pandemic, our focus for this study was to test a heterologous delivery system in the form of membrane-bound stabilized Envs encoded by the Ad26 vector. Moreover, membrane-bound Env has been demonstrated to display a more authentic glycosylation in *Cao et al Nature communications 2018 doi:10.1038/s41467-018-06121-4*, therefore we hypothesized that the Ad26 booster might further drive affinity maturation towards bNAbs. Follow up work would indeed require the assessment of whether a CoPoP boost improves the potency and breadth of the response induced by the three-dose primary CoPoP regimen. Lines 345-348 have now been modified in the text to account for this point. Follow up work is required as a next step to assess whether a CoPoP boost improves the potency and breadth the response induced by the three-dose primary CoPoP regimen.

Major comment #7:

It is unclear why the authors chose the Env-CoPOP2 group and not the Env-CoPOP1 group for heterologous vaccination. The Env-CoPOP1 group appeared to show greater potency and breadth against eight tier 2 clade C strains and other strains on the global panel, as shown in Figure 3a.

Thank you for this comment and we apologize this was not made clear from the text. Since the two groups were similar in a head-to-head comparison of elicited neutralization breadth and mean neutralization titers of the tier 2 panel (summarized in Figure 3b), the choice of CoP2 over CoP1 was based on the improved coupling efficiency of CoP2. The practical consideration of requiring less Env protein to prepare Env-CoPoP

liposomes is one that is expected to aid developability of a vaccine candidate. We discuss this in lines 348-350 in the text to clarify the choice of regimen.

Major comment #8:

The authors reported that “the average breadth coverage of neutralizing responses elicited by the HIV-1 Env-decorated CoPoP particles was up to 35% against a multi-clade panel”. It would make sense to study which epitopes these NAbs target, e.g., using some epitope-knockout mutant viruses.

Thank you very much for this comment. Please also refer to our response to reviewers 1 and 2 on this topic which is as follows: we have performed epitope mapping of the sera using neutralization fingerprinting based on *Georgiev et al Science 2013 doi: 10.1126/science.1233989*. Based on the neutralization fingerprinting data, we see predicted reactivity of the sera from all three groups primarily against the fusion peptide and gp41-gp120 interface. Of note, groups 1 and 2, which showed superior neutralization breadth, display greater reactivity that maps to the variable loops, CD4bs and MPER versus group 3, which could perhaps explain the greater breadth seen in these groups against the global and clade C panels. We have included the data in the manuscript in figure 3c and discussed these observations in lines 213-219 and 326-330.

REVIEWER COMMENTS

Reviewer #1 (Remarks to the Author):

The authors performed neutralization fingerprint based epitope mapping for the week 18 sera to dissect the neutralization specificity. Some suggestions are listed below to help presenting/interpreting the data.

- (1) The method can be more specific regarding the strains used in the assay as well as the reference antibodies that were used;
- (2) The data of the ID50 of each serum for each virus strain should be reported in the supplementary section similar to what has been reported in Fig. 3a;
- (3) In Fig. 3c, please label the key scale, such as "Prediction score" or something like that.
- (4) One major concern is the specificity prediction precision. It is puzzling that the authors had detected MPER-directed neutralization activity (10E8, 2F5-like etc., Fig3c) from many sera, as the immunogens are SOSIP-based designs without MPER region (residue 664 as the last residue at the c-termini). The detection of MPER-directed neutralization specificity was likely caused by assay fluctuations (e.g., impurities from the sera that cause the decrease of virus entry signals non-specifically). This could be the case with other specificities as well. The authors should list the exact prediction scores and be careful with the data interpretation. Perhaps remove the scores of concerns (e.g. background specificity such as the MPER), and only focus on reporting the solid specificities (e.g. the fusion peptide or interface with higher scores) by applying some reasonable cutoff standard based on the confidence for the data.

Reviewer #2 (Remarks to the Author):

The authors have satisfactorily addressed all the points raised in the reviews. Excellent work.

Reviewer #3 (Remarks to the Author):

The authors have addressed my questions and comments in the first review with sufficient details.

Response letter to reviewers

Author responses are in blue

Reviewer #1 (Remarks to the Author):

The authors performed neutralization fingerprint based epitope mapping for the week 18 sera to dissect the neutralization specificity. Some suggestions are listed below to help presenting/interpreting the data.

We thank the reviewer for taking the time to evaluate our work once again and have addressed all additional points raised below.

(1) The method can be more specific regarding the strains used in the assay as well as the reference antibodies that were used;

Thank you for pointing this out. We have now included the information on strains and antibodies used in the methods section at lines 776-782. The antibody names are in addition included in the figure 3c column titles.

(2) The data of the ID50 of each serum for each virus strain should be reported in the supplementary section similar to what has been reported in Fig. 3a;

This is an excellent point and we have now included all the serum ID50 values in a new supplementary table 2.

(3) In Fig. 3c, please label the key scale, such as “Prediction score” or something like that.

Apologies for this. We have now updated the legend scale in the figure 3c to include the label for the predictions scores.

(4) One major concern is the specificity prediction precision. It is puzzling that the authors had detected MPER-directed neutralization activity (10E8, 2F5-like etc., Fig3c) from many sera, as the immunogens are SOSIP-based designs without MPER region (residue 664 as the last residue at the c-termini). The detection of MPER-directed neutralization specificity was likely caused by assay fluctuations (e.g., impurities from the sera that cause the decrease of virus entry signals non-specifically). This could be the case with other specificities as well. The authors should list the exact prediction scores and be careful with the data interpretation. Perhaps remove the scores of concerns (e.g. background specificity such as the MPER), and only focus on reporting the solid specificities (e.g. the fusion peptide or interface with higher scores) by applying some reasonable cutoff standard based on the confidence for the data.

Thank you for raising this point. The immunogens used in the study are indeed repair and stabilized soluble Env antigens which span up to residue 664 (HXb2 amino acid numbering) on the HIV-1 envelope. This includes a few early residues of the bNAb MPER epitopes but not the later parts of the MPER. We have added a new supplementary table 3 that shows the prediction score raw values and have highlighted all values above 0.01 in grey. Based on this, the neutralization fingerprinting predicts serum antibody signatures that resemble 2F5 with a relatively high score and to a lesser extent 10E8.

These observations could be explained when one considers the actual binding footprint of bNAbs on the Env which is normally larger than the key residues that define their epitope specificities. In this instance, we see relatively more 2F5-like signature which can be explained by the presence of residues 657-664 in our immunogens that are contact residues for 2F5 based on structural analysis by *Ofek et al., J. Virology 2004 DOI: 10.1128/JVI.78.19.10724-10737.2004*. Of note, HIV-1 antigens when presented on cholesterol containing liposomes (CoPoP liposomes include cholesterol) have been shown to increase immunogenicity of membrane proximal regions including a dominant site (residues 653-667) that overlaps with the 2F5 binding motif as shown by *Molinos-Albert et al., Scientific reports 2017 DOI: 10.1038/srep40800*. Similarly, 10E8 contact sites on the Env trimer have been shown to span beyond the MPER region and also include upstream gp41 and gp120 regions on the folded trimer as shown in *Lee et al., Science 2016 DOI: 10.1126/science.aad2450*. Therefore, it is possible that based on the predictions of the neutralization fingerprinting method, some antibodies induced by the CoPoP-HIV vaccine regimen may resemble neutralization signatures of the 2F5 and 10E8 MPER antibodies.

With regard to the assay, we always used MLV as an assay control and did not detect any reactivity for this non-HIV-1 virus in this case. Thus, we can conclude that the neutralization patterns observed here are specific and not due to serum impurities causing non-specific signals. Finally, we have now also added a discussion addressing the MPER reactivity in the manuscript text at lines 330-333.

Reviewer #2 (Remarks to the Author):

The authors have satisfactorily addressed all the points raised in the reviews. Excellent work.

Thank you very much for calling our work “excellent” and your time in evaluating our work once again.

Reviewer #3 (Remarks to the Author):

The authors have addressed my questions and comments in the first review with sufficient details.

We thank the reviewer for their time and efforts in evaluating our work and making our study stronger.